

# Geochemical inverse modeling of chemical and isotopic data from groundwaters in Sahara (Ouargla basin, Algeria)

R. Slimani[1], A. Guendouz[2], F. Trolard[3], A. S. Moulla[4], B. Hamdi-Aïssa[1], and G. Bourrié[3]

[1]Univ. Ouargla, Fac. des sciences de la nature et de la vie, Lab. Biochimie des milieux désertiques, Ouargla 30000, Algeria
[2]Blida University, Science and Engineering Faculty, P.O. Box 270 Soumaa, Blida, Algeria
[3]INRA, UMR1114 Emmah, Avignon, France
[4]Algiers Nuclear Research Centre, P.O. Box 399 Alger-RP, Algiers 16000, Algeria

Received: 8 September 2015 – Accepted: 4 October 2015 – Published: 1 February 2015

Correspondence to: R. Slimani (slm_rabia@yahoo.fr)

Published by Copernicus Publications on behalf of the European Geosciences Union.



## Abstract

New samples were collected in the three major Saharan aquifers namely, the "Continental Intercalaire" (CI), the "Complexe Terminal" (CT) and the Phreatic aquifer (Phr) and completed with unpublished more ancient chemical and isotopic data. Instead of classical Debye-Hückel extended law, Specific Interaction Theory (SIT) model, recently incorporated in Phreeqc 3.0 was used. Inverse modeling of hydro chemical data constrained by isotopic data was used here to quantitatively assess the influence of geochemical processes: at depth, the dissolution of salts from the geological formations during upward leakage without evaporation explains the tran sitions from CI to CT and to a first pole of Phr (pole I); near the surface, the dissolution of salts from sebkhas by rainwater explains another pole of Phr (pole II). In every case, secondary precipitation of calcite occurs during dissolution. All Phr waters result from the mixing of these two poles together with calcite precipitation and ion exchange processes. These processes are quantitatively assessed by Phreeqc model. Globally, gypsum dissolution and calcite precipitation were found to act as a carbon sink.

## 1  Introduction

A scientific study published in 2008 showed that 85 % of the world population lives in the driest half of the Earth. More than 1 billion people residing in arid and semi-arid areas of the world have only access to little or no renewable water resources (OECD, 2008). In many arid regions such as Sahara, groundwater is the only source of water supply for domestic, agricultural or industrial purposes, causing most of the time overuse and/or degradation of water quality.

The groundwater resources of Ouargla basin (Lower-Sahara, Algerian; Fig. 1) are contained in three main reservoirs (UNESCO, 1972; Eckstein and Eckstein, 2003; OSS, 2003, 2008):

**HESSD**

doi:10.5194/hess-2015-385

**Chemical and isotopic data from groundwaters in Sahara**

R. Slimani et al.

**HESSD**

doi:10.5194/hess-2015-385

**Chemical and isotopic data from groundwaters in Sahara**

R. Slimani et al.

- at the top, the phreatic aquifer (Phr), located in sandy gypsum permeable formations of Quaternary, is almost unexploited (only north of Ouargla) due to its salinity ($50\,\mathrm{g\,L^{-1}}$);

- in the middle, the "Complexe Terminal" (CT) aquifer, (Cornet and Gouscov, 1952; UNESCO, 1972) which is the most exploited, and includes several aquifers in different geological formations. It circulates in one or two lithostratigraphic formations of the Eocene and Senonian carbonates or Mio-pliocene sands;

- at the bottom, the "Continental Intercalaire" (CI) aquifer, where water is contained in the lower Cretaceous continental formations (Barremian and Albian), mainly composed of sandstones, sands and clays. It is only partially exploited because of its significant depth.

After use, waters are discharged in a closed system (endorheic basin) and constitute a potential hazard to the environment, to public health and may jeopardize the sustainability of agriculture (rising of the phreatic aquifer watertable, extension of soil salinization and so on; Hamdi-Aïssa et al., 2004; Slimani, 2006). Several previous studies (Guendouz, 1985; Fontes et al., 1986; Guendouz and Moulla, 1996; Edmunds et al., 2003; Guendouz et al., 2003; Hamdi-Aïssa et al., 2004; Foster et al., 2006; OSS, 2008; Al-Gamal, 2011) tried, starting from chemical and isotopic information ($^2$H, $^{18}$O, $^{234}$U, $^{238}$U, $^{36}$Cl) to best characterize the relationships between aquifers. They were more specifically tackling the issue of the Continental Intercalaire recharge. These investigations dealt particularly with water chemical facies, mapped isocontents of various parameters, and reported typical geochemical ratios ($[\mathrm{SO_4^{2-}}]/[\mathrm{Cl^-}]$, $[\mathrm{Mg^{2+}}]/[\mathrm{Ca^{2+}}]$) as well as other correlations. Minerals/solutions equilibria were checked by computing saturation indices with respect to calcite, gypsum, anhydrite and halite, but processes were only qualitatively assessed.

In the present study, new data were collected in order to characterize the hydrochemical and the isotopic composition of the major aquifers in Ouargla' region. They also aimed at identifying the origin of the mineralization and water-rock interactions that

occur along the flow. New possibilities offered by progress in geochemical simulations were used. More specifically, Specific Interaction Theory (SIT) recently incorporated in Phreeqc 3.0 (Parkhurst and Appelo, 2013) seems now to bridge the gap between the "extended" Debye-Hückel law that is valid only for dilute solutions and Pitzer's model,
which is mainly used for brines, and which appears now as over parameterized (Grenthe and Plyasunov, 1997). Inverse modeling with Phreeqc 3.0 was used to quantitatively assess the influence of the processes that explain the acquisition of solutes for the different aquifers: dissolution, precipitation, mixing and ion exchange. This results in constraints on mass balances as well as on the exchange of matter between aquifers.

## 2   Methodology

### 2.1   Presentation of the study area

The study area is located in the northeastern desert of Algeria "Lower-Sahara" (Le Houérou, 2009) near the city of Ouargla (Fig. 1), 31°54′ to 32°1′ N and 5°15′ to 5°27′ E, with a mean elevation of 134 (m a.s.l.). It is located in the quaternary fossil
valley of Oued Mya basin. Present climate belongs to the arid Mediterranean-type (Dubief, 1963; Le Houérou, 2009; ONM, 1975/2013). This climate is characterized by a mean annual temperature of 22.5 °C, a yearly rainfall of 43.6 mm yr$^{-1}$ and a very high evaporation rate of 2138 mm yr$^{-1}$.

Geologically, Ouargla's region and the entire Lower Sahara consist of sedimentary
formations (Fig. 2). The basin is carved into Mio-pliocene (MP) deposits, which alternate with red sands, clays and sometimes marls; gypsum is not abundant and dated from Pontian (MP; Cornet and Gouscov, 1952; Dubief, 1953; Ould Baba Sy and Besbes, 2006). The continental Pliocene consists of a local limestone crust with puddingstone or lacustrine limestone (Fig. 2), shaped by æolian erosion into flat areas (regs).
The Quaternary formations are lithologically composed of alternating layers of permeable sand and relatively impermeable marl (Aumassip et al., 1972; Chellat et al., 2014).

## HESSD

doi:10.5194/hess-2015-385

### Chemical and isotopic data from groundwaters in Sahara

R. Slimani et al.

The exploitation of Mio-pliocene aquifer is ancient and at the origin of the creation of the oasis (Lelièvre, 1969; Moulias, 1927). The piezometric level was higher (145 m a.s.l.) but overexploitation at the end of the XIXth century led to a catastrophic decrease of the resource, with presently more than 900 boreholes (ANRH, 2011).

The exploitation of Senonian aquifer dates back to 1953 at a depth 140 to 200 m depth, with a small initial rate ca. 540 L mn$^{-1}$; two boreholes have been exploited since 1965 and 1969, with a total flowrate ca. 2500 L mn$^{-1}$, for drinking water and irrigation.

The exploitation of Albian aquifer dates back to 1956, with a piezometric level 405 m and a pressure 22 kg cm$^{-2}$. Presently, two boreholes are exploited:

– El Hedeb I, 1335 m depth, with a flowrate 141 L s$^{-1}$;

– El Hedeb II, 1400 m depth, with a flowrate 68 L s$^{-1}$.

## 2.2  Sampling and analytical methods

The sampling scheme complies with the flow directions of the two formations (Phr and CT aquifers); for the CI aquifer only five points are available, so it is impossible to choose a transect (Fig. 3). Groundwater samples ($n = 107$) were collected during a field campaign in 2013, along the main flow line of Oued Mya, 67 piezometers tap the phreatic aquifer, 32 wells tap the CT aquifer and 8 boreholes tap the CI aquifer (Fig. 3). Analyses of Na$^+$, K$^+$, Ca$^{2+}$, Mg$^{2+}$, Cl$^-$, SO$_4^{2-}$ and HCO3$^-$ were performed by ion chromatography at Algiers Nuclear Research Center (CRNA). Previous and yet unpublished data (Guendouz and Moulla, 1996) sampled in 1996 are used here too: 59 samples for Phr aquifer, 15 samples for CT aquifer and 3 samples for the CI aquifer for chemical analyses, data $^{18}$O and $^3$H (Guendouz and Moulla, 1996).

## 2.3  Geochemical method

Phreeqc (Parkhurst and Appelo, 2013) was used to check minerals/solution equilibria using the specific interaction theory (SIT), i.e. the extension of Debye-Hückel law by

**HESSD**

doi:10.5194/hess-2015-385

**Chemical and isotopic data from groundwaters in Sahara**

R. Slimani et al.

**HESSD**

doi:10.5194/hess-2015-385

**Chemical and isotopic data from groundwaters in Sahara**

R. Slimani et al.

Scatchard and Guggenheim incorporated recently in Phreeqc 3.0. Inverse modeling was used to calculate the number of minerals and gases' moles that must respectively dissolve or precipitate/degas to account for the difference in composition between initial and final water end members (Plummer and Back, 1980; Kenoyer and Bowser, 1992; Deutsch, 1997; Plummer and Sprinckle, 2001; Guler and Thyne, 2004; Parkhurst and Appelo, 2013). This mass balance technique has been used to quantify reactions controlling water chemistry along flow paths (Thomas et al., 1989). It is also used to quantify the mixing proportions of end-member components in a flow system (Kuells et al., 2000; Belkhiri et al., 2010, 2012).

## 3   Results and discussion

Tables 1 to 5 illustrate the results of the chemical and the isotopic analyses. Samples are ordered according to an increasing salt content that was estimated from their specific electric conductivity (EC). In both phreatic and CT aquifers, temperature is close to 25 °C, while for CI aquifer, temperature is close to 50 °C. The results presented in those tables are raw analytical data that were corrected for defects of charge balance before computing activities with Phreeqc. As analytical errors could not be ascribed to a specific analyte, the correction was made proportionally. The corrections do not affect the anions to anions mole ratios such as for $[HCO3^-]/([Cl^-]+2[SO4^{2-}])$ or $[SO4^{2-}]/[Cl^-]$, whereas they affect the cation to anion ratio such as for $[Na^+]/[Cl^-]$.

### 3.1   Characterization of chemical facies of the groundwater

Piper diagrams drawn for the studied groundwaters (Fig. 4) broadly show a scatter plot dominated by a Chloride-Sodium facies. However, when going into small details, the widespread chemical facies of the Phr aquifer is closer to the NaCl pole than those of CI and CT aquifers. The facies of the Phreatic aquifer most concentrated samples are in the following order: Ca-sulfate < Na-sulfate = Mg-sulfate < Na-chloride. This se-

Interactive Discussion

quential order of solutes is comparable to that of other groundwater occurring in North Africa, and especially in the neighboring area of the chotts (depressions where salts concentrate by evaporation) Merouane and Melrhir (Vallès et al., 1997; Hamdi-Aïssa et al., 2004).

## 3.2 Spatial distribution of the mineralization

The salinity of the phreatic aquifer varies considerably depending on the location (near wells or drains) and time (influence of irrigation; Fig. 5a).

Its salinity is low around irrigated and fairly well-drained areas, such as the palm groves of Hassi Miloud, just north of Ouargla (Fig. 3) that benefit from freshwater and are drained to the sebkha Oum el Raneb. However, the three lowest salinity values are observed in the wells of Ouargla palm-grove itself, where the Phr aquifer watertable is deeper than 2 m.

Conversely, the highest salinity waters are found in wells drilled in the chotts and sebkhas (a sebkha is the central part of a chott where salinity is the largest) Safioune and Oum er Raneb where the aquifer is often shallower than 50 cm.

The salinity of the Complexe Terminal (Mio-pliocene) aquifer (Fig. 5b) is much lower than that of the Phr aquifer, and ranges from 1 to $2\,\mathrm{g\,L^{-1}}$; however, its hardness is larger and it contains more sulfate, chloride and sodium than the waters of the Senonian formations and those of the CI aquifer. The salinity of the Senonian aquifer ranges from 1.1 to $1.7\,\mathrm{g\,L^{-1}}$, while the average salinity of the Continental Intercalaire is $0.7\,\mathrm{g\,L^{-1}}$ (Fig. 5c).

A likely contamination of the Mio-pliocene aquifer by phreatic groundwaters through casing leakage in an area where water is heavily loaded with salt and therefore particularly aggressive cannot be excluded.

**HESSD**

doi:10.5194/hess-2015-385

**Chemical and isotopic data from groundwaters in Sahara**

R. Slimani et al.

Discussion Paper | Discussion Paper | Discussion Paper | Discussion Paper

## 3.3 Saturation Indices

The calculated saturation indices reveal that waters from CI at 50 °C are close to equilibrium with respect to calcite (Figs. 6 top and 7), except for 3 samples that are slightly oversaturated. They are however all undersaturated with respect to gypsum (Figs. 6 bottom and 8).

Moreover, they are oversaturated with respect to dolomite and undersaturated with respect to anhydrite (Fig. 8) and halite (Fig. 9).

Waters from CT and phreatic aquifers show the same pattern, but some of them are more largely oversaturated with respect to calcite, at 25 °C (Fig. 7).

However, several phreatic waters (P031, P566, PLX4, PL18, P002, P023, P116, P066, P162 and P036) that are located in the sebkhas of Sefioune, Oum-er-Raneb, Bamendil and Ain el Beida's chott are saturated with gypsum and anhydrite. This is in accordance with high evaporative environments found elsewhere (UNESCO, 1972; Hamdi-Aïssa et al., 2004; Slimani, 2006).

No significant saturation indices' evolution from the south to the north upstream and downstream of Oued Mya (Fig. 8) is observed. This suggests that the acquisition of mineralization is due to geochemical processes that have already reached equilibrium or steady state in the upstream areas of Ouargla.

## 3.4 Change of facies from the carbonated pole to the evaporites' pole

The facies shifts progressively from the carbonated (CI and CT aquifers) to the evaporites'one (Phr aquifer) with an increase in sulfates and chlorides at the expense of carbonates (SI of gypsum, anhydrite and halite). This is illustrated by a decrease of the following two ratios: $[HCO3^-]/([Cl^-]+2[SO_4^{2-}]$; Fig. 10) from 0.2 to 0 and of the ratio $[SO_4^{2-}]/[Cl^-]$ from 0.8 to values ranging from 0.3 and 0 (Fig. 11) while salinity increases. Carbonate concentrations tend towards very small values, while it is not the case for sulfates. This is due to both gypsum dissolution and calcite precipitation.

**HESSD**

doi:10.5194/hess-2015-385

**Chemical and isotopic data from groundwaters in Sahara**

R. Slimani et al.

Discussion Paper | Discussion Paper | Discussion Paper | Discussion Paper

**HESSD**

doi:10.5194/hess-2015-385

**Chemical and isotopic data from groundwaters in Sahara**

R. Slimani et al.

Chlorides in groundwater may come from three different sources: (i) ancient sea water entrapped in sediments; (ii) dissolution of halite and related minerals that are present in evaporite deposits and (iii) dissolution of dry fallout from the atmosphere, particularly in these arid regions (Matiatos et al., 2014; Hadj-Ammar et al., 2014).

For most of the sampled points the $[Na^+]/[Cl^-]$ ratio remains close to 1, but significant ranges are observed: from 0.85 to 1.26 for CI aquifer, from 0.40 to 1.02 for the CT aquifer and from 0.13 to 2.15 for the Phr aquifer. All the measured points from the three considered aquifers are more or less linearly scattered around the unity slope straight line that stands for halite dissolution (Fig. 12). The latter appears as the most dominant reaction occurring in the medium. However, at very high salinity, $Na^+$ seems to swerve from the straight line, towards smaller values.

A further scrutiny of (Fig. 12) shows that CI waters are very close to the 1 : 1 line. CT waters are enriched in both $Na^+$ and $Cl^-$ but slightly lower than the 1 : 1 line while phreatic waters are largely enriched and much more scattered. CT waters are closer to the seawater mole ratio (0.858), but some lower values imply a contribution from another source of chloride than halite or from entrapped seawater. Conversely, a $[Na^+]/[Cl^-]$ ratio larger than 1 is observed for phreatic waters, which implies the contribution of another source of sodium, most likely sodium sulfate, that is present as mirabilite or thenardite in the chotts and the sebkhas areas.

$[Br^-]/[Cl^-]$ ratio ranges from $2 \times 10^{-3}$ to $3 \times 10^{-3}$. The value of this molar ratio for halite is around $2.5 \times 10^{-3}$, which matches the aforementioned range and confirms that halite dissolution is the most dominant reaction taking place in the studied medium.

In these aquifers, calcium originates both from carbonate and sulfate (Figs. 13 and 14). Three samples from CI aquifer are close to the $[Ca^{2+}]/[HCO3^-]$ 1 : 2 line, while calcium sulfate dissolution explains the excess of calcium. However, a small but significant number of samples (9) from phreatic aquifer are depleted in calcium, and plot under the $[Ca^{2+}]/[HCO3^-]$ 1:2 line. This cannot be explained by precipitation of calcite, as some are undersaturated with respect to that mineral, while others are oversaturated.

**HESSD**

doi:10.5194/hess-2015-385

**Chemical and isotopic data from groundwaters in Sahara**

R. Slimani et al.

In this case, a cation exchange process seems to occur leading to a preferential adsorption of divalent cations, with a release of Na$^+$. This is confirmed by the inverse modeling that is developed below and which implies Mg$^{2+}$ fixation and Na+ and K+ releases.

Larger sulfate values observed in the phreatic aquifer (Fig. 14) with $[Ca^{2+}]/[SO4^{2-}]<1$ can be attributed to a sodium-magnesium sulfate dissolution from a mineral bearing such elements. This is for instance the case of bloedite.

### 3.5  Isotope geochemistry

CT and CI aquifer exhibit depleted and homogeneous $^{18}$O contents, ranging from $-8.32$
to $-7.85$‰. This was already previously reported by many authors (Edmunds et al., 2003; Guendouz et al., 2003; Moulla et al., 2012). On the other hand, $^{18}$O values for the phreatic aquifer are widely dispersed and vary between $-8.84$ to $3.42$‰ (Table 6). Waters located north of the Hassi Miloud to Sebkhet Safioune axis are more enriched in heavy isotopes and therefore more evaporated. In that area, water table is
close to the surface and mixing of both CI and CT groundwaters with phreatic ones through irrigation is nonexistent. Conversely, waters located south of Hassi Miloud up to Ouargla city show depleted values. This is the clear fingerprint of a contribution to the Phr waters from the underlying CI and CT aquifers (Gonfiantini et al., 1975; Guendouz, 1985; Fontes et al., 1986; Guendouz and Moulla, 1996).
Phreatic waters result from a mixing of two end-members. An evidence for this is given by considering the ([Cl$^-$], $^{18}$O) relationship (Fig. 15). The two poles are: (i) a first pole of $^{18}$O depleted groundwater (Fig. 16), and (ii) another pole of $^{18}$O enriched groundwater with positive values and a high salinity. The latter is composed of phreatic waters occurring in the northern part of the study region.
Pole I represents the waters from CI and CT whose isotopic composition is depleted in $^{18}$O (average value around $-8.2$‰; Fig. 15). They correspond to an old water recharge (palæorecharge); whose age estimated by means of $^{14}$C, exceeds 15 000 years BP (Guendouz, 1985; Guendouz and Michelot, 2006). So, it is not a water

body that is recharged by recent precipitation. It consists of CI and CT groundwaters and partly of phreatic waters, and can be ascribed to an upward leakage favored by the extension of faults near Amguid El-Biod dorsal.

Pole II, observed in Sebkhet Safioune, can be ascribed to the direct dissolution of surficial evaporitic deposits conveyed by evaporated rainwater.

Evaporation alone cannot explain the distribution of data that is observed (Fig. 15). An evidence for this is given in a semi-logarithmic plot (Fig. 16), as classically obtained according to the simple approximation of Rayleigh equation (cf. Appendix):

$$\delta^{18}O \approx 1000 \times (1 - \alpha)\log[Cl-] + cte, \tag{1}$$

$$\approx -\epsilon \log[Cl-] + cte, \tag{2}$$

where $\alpha$ is the fractionation factor during evaporation, and $\epsilon \equiv -1000 \times (1 - \alpha)$ is the enrichment factor, and cte is an abbreviation for constant (Ma et al., 2010; Chkir et al., 2009).

CI and CT waters are better separated in the semi-logarithmic plot because they are differentiated by their chloride content. According to Eq. (1), simple evaporation gives a straight line (solid line in Fig. 16). The value of $\epsilon$ used is the value at 25 °C, which is equal to −73.5. There is only one sample (P115) on the evaporation straight line, which could be considered as an outlier in Fig. 15 ([Cl−] $\simeq$ 0). All other samples fit on the logarithmic curve derived from the mixing line illustrated by Fig. 15.

The phreatic waters that are close to pole I (Fig. 15) correspond to groundwaters occurring in the edges of the basin (Hassi Miloud, piezometer P433; Fig. 16). They are low-mineralized and acquire their salinity via two processes namely: dissolution of evaporites along their underground transit up to Sebkhet Safioune and dilution through upward leakage by the less-mineralized waters of CI and CT aquifers (for example Hedeb I for CI and D7F4 for CT; Fig. 16; Guendouz, 1985; Guendouz and Moulla, 1996).

**HESSD**

doi:10.5194/hess-2015-385

**Chemical and isotopic data from groundwaters in Sahara**

R. Slimani et al.

Discussion Paper | Discussion Paper | Discussion Paper | Discussion Paper

The rates of the mixing that are due to upward leakage from CI to CT towards the phreatic aquifer can be calculated by means of a mass balance equation. It only requires knowing the $\delta$ values of each fraction that is involved in the mixing process.

The $\delta$ value of the mixture is given by:

$$\delta_{\text{mix}} = f_1 \times \delta_1 + f_2 \times \delta_2 \tag{3}$$

where $f_1$ is the fraction of CI aquifer, $f_2$ the fraction of the CT and $\delta_1$, $\delta_2$ are the respective isotope contents.

Average values of mixing fractions from each aquifer to the phreatic waters computed by means of Eq. (3) gave the rates of 65 % for CI aquifer and 35 % for CT aquifer.

A mixture of a phreatic water component that is close to pole I (i.e. P433) with another one which is rather close to pole II (i.e. P039; Figs. 15 and 16), for an intermediate water with a $\delta^{18}$O signature ranging from −5 to −2‰ gives mixture fraction values of 52 % for pole I and 48 % for pole II. Isotope results will be used to independently cross-check the validity of the mixing fractions derived from an inverse modeling involving chemical data (cf. infra).

Turonian evaporites are found to lie in between CI deep aquifer, and the Senonian and Miocene formations bearing CT aquifer. CT waters can thus simply originate from ascending CI waters that dissolve Turonian evaporites, a process which does not involve any change in $^{18}$O content. Conversely, phreatic waters result to a minor degree from evaporation, and mostly from dissolution of sebkhas evaporites by $^{18}$O enriched rainwater and mixing with CI-CT waters.

### 3.5.1 Tritium content of water

Tritium contents of Phr aquifer are relatively small (Table 6), they vary between 0 and 8 TU. Piezometers PZ12, P036 and P068 show values close to 8 TU, piezometers P018, P019, P416, P034, P042 and P093 exhibit values ranging between 5 and 6 TU, and the rest of the samples' concentrations are lower than 2 TU. The comparison of

**HESSD**

doi:10.5194/hess-2015-385

**Chemical and isotopic data from groundwaters in Sahara**

R. Slimani et al.

Discussion Paper | Discussion Paper | Discussion Paper | Discussion Paper

these results with that of precipitation which was 16 TU in 1992 suggests the existence of a mixture of water infiltrated before 1950 and a more recent one corresponding to the 1980s (Guendouz and Moulla, 1996; Edmunds et al., 2003; Guendouz et al., 2003; Moulla et al., 2012; ONM, 1975/2013). This is in agreement with the recorded hydro-chemical and stable isotope data.

## 3.6 Inverse modeling

We assume that the relationship between $^{18}$O and Cl− data obtained in 1996 is stable with time, which is a logical assumption as times of transfer from CI to both CT and Phr are very long. Considering both $^{18}$O and Cl− data, thus CI, CT and Phr data populations can be categorized as follows:

- CI does not show appreciable $^{18}$O variations. Its data can be considered as a single population;

- the same holds for CT;

- Phr samples consist however of different populations:

  - pole I, with $\delta^{18}$O values close to -8, and small Cl$^-$ concentrations, more specifically less than 35 mmol L$^{-1}$ (Fig. 17);

  - pole II, with $\delta^{18}$O values larger than 3, and very large Cl$^-$ concentrations, more specifically larger than 4000 mmol L$^{-1}$ (Fig. 17);

  - intermediate Phr samples resulting from mixing between poles I and II (mixing line in Fig. 15, mixing curve in Fig. 16);

  - intermediate samples resulting from evaporation of pole I (evaporation line in Fig. 16).

Statistical parameters for CI, CT, Phr pole I and Phr pole II are given in Table 7.

**HESSD**

doi:10.5194/hess-2015-385

**Chemical and isotopic data from groundwaters in Sahara**

R. Slimani et al.

Discussion Paper | Discussion Paper | Discussion Paper | Discussion Paper

The mass-balance modeling has shown that relatively few phases are required to derive observed changes in water chemistry and to account for the hydrochemical evolution in Ouargla's region. The mineral phases' selection is based upon geological descriptions and analysis of rocks and sediments from the area (OSS, 2003; Hamdi-Aïssa et al., 2004).

The inverse model was constrained so that mineral phases from evaporites including gypsum, halite, mirabilite, glauberite, sylvite and bloedite were set to dissolve until they reach saturation, and calcite, dolomite were set to precipitate once they reached saturation. Cation exchange reactions of $Ca^{2+}$, $Mg^{2+}$, $K^+$ and $Na^+$ on exchange sites were included in the model to check which cations are adsorbed or desorbed during the process. Dissolution and desorption contribute as positive terms in the mass balance, as elements are released in solution. On the other hand, precipitation and adsorption contribute as negative terms, while elements removed from the solution. $CO_{2_{(g)}}$ dissolution is considered by Phreeqc as a dissolution of a mineral, whereas $CO_{2_{(g)}}$ degassing is dealt with as if it were a mineral precipitation.

Inverse modelling leads to a quantitative assessment of the different solutes' acquisition processes and a mass balance for the salts that are dissolved or precipitated from CI, CT and Phr groundwaters (Fig. 16, Table 8), as follows:

– transition from CI to CT involves gypsum, halite and sylvite dissolution, and some ion exchange namely calcium and potassium fixation on exchange sites against magnesium release, with a very small and quite negligible amount of $CO_{2_{(g)}}$ degassing. The maximum elemental concentration fractional error equals 1 %. The model consists of a minimum number of phases (i.e. 6 solid phases and $CO_{2_{(g)}}$); Another model implies as well dolomite precipitation with the same fractional error;

– transition from CT to an average water component of pole I involves dissolution of halite, sylvite, and bloedite from Turonian evaporites, with a very tiny calcite precipitation. The maximum fractional error in elemental concentration is 4 %. An-

**HESSD**

doi:10.5194/hess-2015-385

**Chemical and isotopic data from groundwaters in Sahara**

R. Slimani et al.

Discussion Paper | Discussion Paper | Discussion Paper | Discussion Paper

other model implies $CO_{2(g)}$ escape from the solution, with the same fractional error. Large amounts of $Mg^{2+}$ and $SO_4^{2-}$ are released within the solution (Sharif et al., 2008; Li et al., 2010; Carucci et al., 2012);

– the formation of Phr pole II can be modeled as being a direct dissolution of salts from the sebkha by rainwater with positive $\delta^{18}O$; the most concentrated water (P036 from Sebkhet Safioune) is taken here for pole II, and pure water as rainwater. In a decreasing order of amounts respectively involved in that process, halite, sylvite, gypsum and huntite dissolve, and little calcite precipitates while some $Mg^{2+}$ are released vs. K+ fixation on exchange sites. The maximum elemental fractional error in the concentration is equal to 0.004 %. Another model implies dolomite precipitation with some more huntite dissolving, instead of calcite precipitation, but salt dissolution and ion exchange are the same. Huntite, dolomite and calcite stoichiometries are linearly related, so both models can fit field data, but calcite precipitation is preferred compared to dolomite precipitation at low temperature;

– the origin of all phreatic waters can be explained by a mixing in variable proportions of pole I and pole II. For instance, waters from pole I and pole II can easily be separated by their $\delta^{18}O$ respectively close to $-8$ and $+3.5$ ‰ (Figs. 15 and 16). Mixing the two poles is of course not an inert reaction, but rather results in the dissolution and the precipitation of minerals. Inverse modeling is then used to compute both mixing rates and the extent of matter exchange between soil and solution. For example, a phreatic water (piezometer P068) with intermediate values ($\delta^{18}O = -3$ and $[Cl^-] \simeq 2M$) is explained by the mixing of 58 % water from pole I and 42 % from pole II. In addition, calcite precipitates, $Mg^{2+}$ fixes on exchange sites, against $Na^+$ and $K^+$, gypsum dissolves as well as a minor amount of huntite (Table 8). The maximum elemental concentration fractional error is 2.5 % and the mixing fractions' weighted the $\delta^{18}O$ is $-3.17$ ‰, which is is very close to the measured value ($-3.04$ ‰). All the other models, making use

Discussion Paper | Discussion Paper | Discussion Paper | Discussion Paper | Discussion Paper |

**HESSD**

doi:10.5194/hess-2015-385

**Chemical and isotopic data from groundwaters in Sahara**

R. Slimani et al.

of a minimum number of phases, and not taking into consideration ion exchange reactions are not found compatible with isotope data. Mixing rates obtained with such models are for example 98 % of pole I and 0.9 % of pole II, which leads to a $\delta^{18}O = (-7.80\,‰)$ which is quite far for the real measured value ($-3.04\,‰$).

The main types of groundwaters occurring in Ouargla basin are thus explained and could quantitatively be reconstructed. An exception is however sample P115, which is located exactly on the evaporation line of Phr pole I. Despite numerous attempts, it could not be quantitatively rebuilt. Its $^3$H value (6.8) indicates that it is derived from a more or less recent water component with very small salt content, most possibly affected by rainwater and some preferential flow within the piezometer. As this is the only sample on this evaporation line, there remains a doubt on its significance.

Globally, the summary of mass transfer reactions occurring in the studied system (Table 8) shows that gypsum dissolution results in calcite precipitation and $CO_{2(g)}$ dissolution, thus acting as an inorganic carbon sink.

## 4   Conclusions

Groundwater hydrochemistry is a good record indicator for the water-rock interactions that occur along the groundwater flowpath. The mineral load reflects well the complex processes taking place while water circulates underground since its point of infiltration.

The hydrochemical study of the aquifer system occurring in Ouargla's basin allowed us to identify the origin of its mineralization. Waters exhibit two different facies: sodium chloride and sodium sulfate for the phreatic aquifer (Phr), sodium sulfate for the Complexe Terminal (CT) aquifer and sodium chloride for the Continental Intercalaire (CI) aquifer. Calcium carbonate precipitation and evaporite dissolution explain the facies change from carbonate to sodium chloride or sodium sulfate. However reactions imply many minerals with common ions, deep reactions without evaporation as well as shallow processes affected by both evaporation and mixing. Those processes are sep-

**HESSD**

doi:10.5194/hess-2015-385

**Chemical and isotopic data from groundwaters in Sahara**

R. Slimani et al.



arated by considering both chemical and isotopic data, and quantitatively explained making use of an inverse geochemical modeling.

The main result is that Phr waters do not originate simply from infiltration of rainwater and dissolution of salts from the sebkhas. Conversely, Phr waters are largely influenced by the upwardly mobile deep CT and CI groundwaters, fractions of the latter interacting with evaporites from Turonian formations. Phreatic waters occurrence is explained as a mixing of two end-member components: pole I, which is very close to CI and CT, and pole II, which is highly mineralized and results from the dissolution by rainwater of salts from the sebkhas.

At depth, CI leaks upwardly and dissolves gypsum, halite and sylvite, with some ion exchange, to give waters of CT aquifer composition.

CT transformation into Phr pole I waters involves the dissolution of Turonian evaporites (halite, sylvite and bloedite) with minor calcite precipitation.

At the surface, direct dissolution by rainwater of salts from sebkhas (halite, sylvite, gypsum and some huntite) with precipitation of calcite and $Mg^{2+}/K^+$ ion exchange results in pole II Phr composition.

All phreatic groundwaters result from a mixing of pole I and pole II water that is accompanied by calcite precipitation, fixation of $Mg^{2+}$ on ion exchange sites against the release of $K^+$ and $Na^+$.

Moreover, some $CO_{2(g)}$ escapes from the solution at depth, but dissolves much more at the surface. The most complex phenomena occur during the dissolution of Turonian evaporites while CI leaks upwardly towards CT, and from Phr I to Phr II, while the transition from CT to Phr I implies a very limited number of phases. Globally, gypsum dissolution and calcite precipitation processes both act as an inorganic carbon sink.

Discussion Paper | Discussion Paper | Discussion Paper | Discussion Paper |

**HESSD**

doi:10.5194/hess-2015-385

**Chemical and isotopic data from groundwaters in Sahara**

R. Slimani et al.

**HESSD**

doi:10.5194/hess-2015-385

**Chemical and isotopic data from groundwaters in Sahara**

R. Slimani et al.

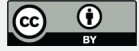

## Appendix A

According to a simple Rayleigh equation, the evolution of the heavy isotope ratio in the remaining liquid $R_l$ is given by:

$$R_l \approx R_{l,0} \times f_l^{\alpha-1}, \tag{A1}$$

where $f_l$ is the fraction remaining liquid and $\alpha$ the fractionation factor.

The fraction remaining liquid is derived from chloride concentration, as chloride can be considered as conservative during evaporation: all phreatic waters are undersaturated with respect to halite, that precipitates only in the last stage. Hence, the following equation holds:

$$f_l \equiv \frac{n_{w,1}}{n_{w,0}} = \frac{[Cl^-]_0}{[Cl^-]_1}. \tag{A2}$$

By taking natural logarithms, one obtains:

$$\ln R_l \approx (1-\alpha) \times \ln[Cl^-] + cte, \tag{A3}$$

As, by definition,

$$R_l \equiv R_{std.} \times (1 + \frac{\delta^{18}O}{1000}), \tag{A4}$$

one has:

$$\ln R_l \equiv \ln R_{std.} + \ln(1 + \frac{\delta^{18}O}{1000}), \tag{A5}$$

$$\approx \ln R_{std.} + \frac{\delta^{18}O}{1000}, \tag{A6}$$

hence, with base 10 logarithms:

$$\delta^{18}O \approx 1000(1 - \alpha)\log[Cl^-] + cte, \qquad (A7)$$

$$\approx -\epsilon \log[Cl^-] + cte, \qquad (A8)$$

where as classically defined $\epsilon = 100(\alpha - 1)$ is the enrichment factor.

*Acknowledgements.* The authors wish to thank the staff members of the National Agency for Water Resources in Ouargla (ANRH) and the Laboratory of Algerian waters (ADE) for the support provided to the Technical Cooperation within which this work was carried out. Analyses of $^{18}$O were funded by the project CDTN/DDHI (Guendouz and Moulla, 1996). The supports of University of Ouargla and of INRA for travel grants of R. Slimani and G. Bourrié are gratefully
acknowledged too.

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

## HESSD

doi:10.5194/hess-2015-385

**Chemical and isotopic data from groundwaters in Sahara**

R. Slimani et al.

**Table 1.** Field and analytical data for the Continental Intercalaire aquifer.

| Locality | Lat. | Long. | Elev. | Date | EC | $t$ | pH | Alk. | Cl$^-$ | SO$_4^{2-}$ | Na$^+$ | K$^+$ | Mg$^{2+}$ | Ca$^{2+}$ | Br$^-$ |
|---|---|---|---|---|---|---|---|---|---|---|---|---|---|---|---|
| | | m | | | mS cm$^{-1}$ | °C | | | | | mmol L$^{-1}$ | | | | |
| Hedeb I | 3534750 | 723986 | 134.8 | 09/11/2012 | 2.01 | 46.5 | 7.65 | 3.5 | 5.8 | 6.79 | 10.7 | 0.63 | 2.49 | 3.3 | 0.034 |
| Hedeb I | 3534750 | 723986 | 134.8 | 1996 | 1.9 | 49.3 | 7.35 | 0.42 | 5.81 | 1.07 | 5.71 | 0.18 | 0.77 | 0.48 | |
| Hadeb II | 3534310 | 724290 | 146.2 | 1996 | 2.02 | 47.4 | 7.64 | 0.58 | 6.19 | 1.22 | 5.06 | 0.2 | 1.28 | 0.82 | |
| Aouinet Moussa | 3548896 | 721076 | 132.6 | 1996 | 2.2 | 48.9 | 7.55 | 1.28 | 6.49 | 1.28 | 5.65 | 0.16 | 1.14 | 1.17 | |
| Aouinet Moussa | 3548896 | 721076 | 132.6 | 22/02/2013 | 2.2 | 48.9 | 7.55 | 3.19 | 9.8 | 3.89 | 6.3 | 0.69 | 5.71 | 1.27 | |
| Hedeb I | 3534750 | 723986 | 134.8 | 11/12/2010 | 2.19 | 49.3 | 7.35 | 1.91 | 12.4 | 4.58 | 10.7 | 0.7 | 3.77 | 2.35 | |
| Hadeb II | 3534310 | 724290 | 146.2 | 11/12/2010 | 2.26 | 47.4 | 7.64 | 2.11 | 13.1 | 5.46 | 13.9 | 0.53 | 4.53 | 1.41 | |
| Hassi Khfif | 3591659.8 | 721636.5 | 110 | 24/02/2013 | 2.43 | 50.5 | 6.83 | 2.98 | 14.3 | 5.24 | 10.8 | 0.84 | 3.44 | 4.63 | 0.033 |
| Hedeb I | 3534750 | 723986 | 134.8 | 27/02/2013 | 2.01 | 46.5 | 7.65 | 3.46 | 15.1 | 7.67 | 11.8 | 0.51 | 5.57 | 5.16 | |
| Hassi Khfif | 3591659.8 | 721636.5 | 110 | 09/11/2012 | 2 | 50.1 | 7.56 | 3.31 | 15.3 | 7.77 | 12.2 | 0.59 | 5.77 | 4.95 | |
| El-Bour | 3560264 | 720366 | 160 | 22/02/2013 | 2.96 | 54.5 | 7.34 | 2.58 | 18.6 | 6.21 | 20.6 | 0.66 | 4.79 | 1.38 | |

**Table 2.** Field and analytical data for the Complexe Terminal aquifer.

| Locality | Site | Aquifer | Lat. | Long. | Elev. | Date | EC | t | pH | Alk. | Cl− | SO₄²⁻ | Na⁺ | K⁺ | Mg²⁺ | Ca²⁺ | Br⁻ |
|---|---|---|---|---|---|---|---|---|---|---|---|---|---|---|---|---|---|
| | | | | m | | | mS cm$^{-1}$ | °C | | | | | mmol L$^{-1}$ | | | | |
| Bamendil | D7F4 | M | 3560759.6 | 720586.2 | 296 | 20/01/2013 | 2.02 | 20.1 | 7.86 | 1.63 | 10.1 | 5.79 | 9.88 | 0.68 | 3.92 | 2.51 | |
| Bamendil | D7F4 | M | 3560759.6 | 720586.2 | 296 | 1996 | 2 | 21.1 | 8.2 | 0.96 | 10.6 | 3.54 | 10.61 | 0.09 | 2.33 | 1.8 | |
| Ifri | D1F151 | S | 3538891.7 | 721060.5 | 204 | 1996 | 2.67 | 23.5 | 7 | 1.26 | 10.75 | 2.71 | 7.99 | 0.73 | 2.32 | 2.12 | |
| Said Otba | D2F66 | S | 3540257.3 | 720085.4 | 216 | 1996 | 2.31 | 24 | 8 | 1.43 | 11.02 | 4.73 | 11.47 | 0.16 | 2.07 | 3.33 | |
| Oglat Larbaâ | D6F64 | M | 3566501.4 | 729369.3 | 177 | 1996 | 2.31 | 18 | 7.9 | 1.41 | 11.36 | 6.55 | 11.59 | 2.31 | 1.96 | 4.58 | |
| El-Bour | D4F94 | M | 3536245.2 | 722641.7 | 100.6 | 27/01/2013 | 3.05 | 26.2 | 7.37 | 1.61 | 12.8 | 6.79 | 5.15 | 1.94 | 1.65 | 9.13 | |
| Said Otba I | D2F71 | S | 3557412.4 | 718272.8 | 211.9 | 1996 | 2.27 | 24.2 | 8.2 | 1.54 | 13.53 | 5.72 | 14.99 | 0.33 | 3.28 | 2.57 | |
| Debiche | D6F61 | M | 3547557.1 | 717067.1 | 173.5 | 26/01/2013 | 2.22 | 23.9 | 7.74 | 1.78 | 14.2 | 4.41 | 12.6 | 0.66 | 5.38 | 4.43 | |
| Rouissat III | D3F10 | S | 3535068.1 | 722352.1 | 248 | 1996 | 3.1 | 26.1 | 7.27 | 2.39 | 14.27 | 6.89 | 13.05 | 0.4 | 3.36 | 5.42 | |
| Said Otba I | D2F71 | S | 3557412.4 | 718272.8 | 211.9 | 26/01/2013 | 5.63 | 25.1 | 7.34 | 2.38 | 14.3 | 6.86 | 13.1 | 0.4 | 3.36 | 5.43 | 0.034 |
| Rouissat III | D3F10 | S | 3535068.1 | 722352.1 | 248 | 20/01/2013 | 2.32 | 18.9 | 7.98 | 1.65 | 15.2 | 8.64 | 12.6 | 1.56 | 5.79 | 4.25 | |
| Ifri | D1F151 | S | 3538891.7 | 721060.5 | 204 | 27/01/2013 | 2.37 | 22.9 | 7.79 | 1.75 | 15.4 | 8.31 | 13.7 | 0.22 | 5.17 | 4.75 | |
| Said Otba | D2F66 | S | 3540257.3 | 720085.4 | 216 | 31/01/2013 | 2.38 | 24.9 | 7.91 | 2.19 | 16.1 | 8.65 | 16.5 | 0.74 | 4.93 | 4.29 | |
| Oglat Larbaâ | D6F64 | M | 3566501.4 | 729369.3 | 177 | 31/01/2013 | 2.43 | 23.7 | 7.62 | 2.3 | 16.3 | 8.65 | 13.6 | 0.71 | 5.86 | 4.97 | |
| SAR Mekhadma | D1F91 | S | 3536757.7 | 717822.3 | 221 | 03/02/2013 | 2.47 | 25.8 | 7.75 | 3.43 | 16.5 | 8.53 | 16.1 | 0.68 | 5.27 | 4.92 | |
| Sidi Kouiled | D9F12 | S | 3540855.1 | 729055.4 | 329 | 24/01/2013 | 2.57 | 21.3 | 8.05 | 4.65 | 16.8 | 8.85 | 16.1 | 0.79 | 6.21 | 5.01 | |
| Ain N'sara | D6F50 | S | 3559323.6 | 716868.4 | 255 | 25/01/2013 | 3.36 | 25.7 | 7.36 | 1.98 | 16.9 | 9.71 | 15.9 | 0.35 | 3.39 | 7.87 | 0.033 |
| A.Louise | D4F73 | S | 3537523.4 | 721904.6 | 310 | 1996 | 2.57 | 24 | 7.49 | 1.98 | 17.4 | 9.04 | 13.9 | 1.99 | 5.78 | 5.05 | |
| Ghazalet A.H | D6F79 | M | 3598750.2 | 720356.8 | 119 | 02/02/2013 | 2.84 | 22.5 | 7.55 | 3.47 | 17.4 | 9.35 | 16.6 | 0.62 | 6.24 | 4.96 | |
| Ain moussa II | D9F30 | S | 3537814.1 | 719665.1 | 220.6 | 02/02/2013 | 7.52 | 23.9 | 7.52 | 2.37 | 17.5 | 8.24 | 17.3 | 0.39 | 3.1 | 6.46 | 0.033 |
| Ain N'sara | D6F50 | S | 3559323.6 | 716868.4 | 255 | 02/02/2013 | 2.62 | 23.9 | 7.65 | 2.11 | 17.7 | 9.19 | 15.5 | 1.13 | 6.11 | 4.73 | |
| H.Miloud | D1F135 | M | 3547557.1 | 717067.1 | 173 | 03/02/2013 | 2.76 | 21.6 | 7.55 | 3.32 | 17.9 | 9.22 | 16.5 | 1.01 | 6.17 | 4.91 | |
| El Bour | D6F97 | S | 3540936.5 | 715816.0 | 169 | 25/01/2013 | 2.65 | 19.9 | 8.02 | 2.14 | 17.9 | 9.28 | 15.8 | 1.6 | 5.84 | 4.73 | |
| H.Miloud | D1F135 | M | 3547557.1 | 717067.1 | 173 | 1996 | 2.07 | 22.7 | 8.1 | 2.8 | 18.08 | 5.73 | 16.61 | 0.51 | 3.65 | 4.26 | |
| N'goussa El Hou | D6F51 | S | 3556256.7 | 718979.5 | 198 | 31/01/2013 | 2.97 | 22.9 | 7.52 | 2.03 | 18.4 | 9.63 | 17.1 | 0.45 | 6.17 | 4.99 | |
| El Koum | D6F67 | S | 3573694.1 | 721639.7 | 143 | 21/01/2013 | 3.07 | 22.9 | 8.09 | 3.52 | 18.4 | 9.71 | 17.9 | 0.32 | 6.49 | 5.14 | |
| El Koum | D6F67 | S | 3573694.1 | 721639.7 | 143 | 1996 | 2.5 | 25 | 7.6 | 1.5 | 18.79 | 7.17 | 10.18 | 3.43 | 4.97 | 5.81 | |
| ITAS | D1F150 | M | 3536186.6 | 717046.1 | 93.1 | 21/01/2013 | 3.66 | 23.9 | 7.54 | 1.48 | 18.8 | 7.07 | 10.1 | 3.41 | 4.94 | 5.77 | |
| Ain moussa V | D9F13 | M | 3538409.2 | 718680.2 | 210.2 | 08/02/2013 | 2.39 | 25.3 | 7.22 | 2.28 | 19.4 | 9.45 | 18.8 | 0.39 | 3.31 | 7.61 | 0.034 |
| El-Bour | D4F94 | M | 3536245.2 | 722641.7 | 100.6 | 1996 | 2.3 | 21.2 | 7.9 | 1.58 | 20.05 | 7.21 | 12.09 | 2.62 | 5.76 | 5.17 | |
| Rouissat I | D3F18 | M | 3535564.2 | 722498.9 | 80.4 | 26/01/2013 | 3.13 | 23 | 8.1 | 3.15 | 21.2 | 11.1 | 19.6 | 0.87 | 7.08 | 6.01 | |
| Rouissat I | D3F18 | M | 3535564.2 | 722498.9 | 80.4 | 1996 | 2 | 20 | 7.84 | 1.86 | 21.66 | 8.46 | 17.72 | 1.19 | 5.05 | 6 | |
| St. pompage chott | D5F80 | S | 3541656.9 | 723521.9 | 224.1 | 04/02/2013 | 3.28 | 24.5 | 8.23 | 3.91 | 22.1 | 11.9 | 19.9 | 2.13 | 7.64 | 6.28 | |
| Chott Palmeraie | D5F77 | S | 3538219.3 | 725541.3 | 242.8 | 05/02/2013 | 3.37 | 24.6 | 7.53 | 3.26 | 22.3 | 12.1 | 20.9 | 1.15 | 8.25 | 5.78 | |
| Bour El Haicha | D1F134 | M | 3545533.1 | 720391.7 | 86 | 05/02/2013 | 3.4 | 22.2 | 7.34 | 4.13 | 23.2 | 12.2 | 21.2 | 1.49 | 8.61 | 6.01 | |
| Abazat | D2F69 | M | 3552504.9 | 712786.3 | 137.1 | 03/02/2013 | 3.54 | 24.6 | 7.61 | 2.24 | 24.7 | 12.7 | 21.1 | 1.65 | 8.45 | 6.47 | |
| Garet Chemia | D1F113 | S | 3536174.1 | 716808.5 | 213.7 | 28/01/2013 | 4.05 | 28 | 7.3 | 2.21 | 25.9 | 9.47 | 25.4 | 0.57 | 3.64 | 7.17 | 0.037 |
| Frane | D6F62 | M | 3570175.8 | 717133.8 | 167.5 | 27/01/2013 | 3.79 | 24.2 | 7.95 | 2.27 | 25.9 | 13.5 | 22.6 | 0.64 | 8.91 | 7.16 | |
| Oum Raneb | D6 F69 | M | 3540451.1 | 721919.8 | 215.8 | 25/01/2013 | 4.2 | 24.1 | 7.03 | 2.61 | 27.9 | 8.67 | 22.9 | 0.62 | 4.42 | 7.96 | 0.035 |
| N'goussa El Hou | D6F51 | S | 3556256.7 | 718979.5 | 198 | 1996 | 3.15 | 23.2 | 8 | 2.59 | 28.39 | 8.61 | 23.14 | 0.62 | 4.46 | 8.01 | |
| H.Miloud Benyaza | D1F138 | M | 3551192.5 | 717042.1 | 88.9 | 28/01/2013 | 3.85 | 25.2 | 7.61 | 2.44 | 28.4 | 14.2 | 23.9 | 1.66 | 10.01 | 7.12 | |
| Ain Laarab | D6F49 | M | 3558822.6 | 716799.1 | 156.5 | 28/01/2013 | 3.97 | 23.7 | 7.33 | 2.16 | 28.9 | 9.01 | 23.9 | 0.53 | 5 | 7.72 | 0.037 |
| H.Miloud Benyaza | D1F138 | M | 3551192.5 | 717042.1 | 88.9 | 1996 | 2.9 | 22.8 | 7.5 | 2.16 | 28.92 | 9.03 | 23.87 | 0.52 | 4.99 | 7.7 | |
| Rouissat | D3F8 | M | 3545470.7 | 732837.6 | 332.4 | 03/02/2013 | 4.38 | 25.4 | 7.51 | 1.71 | 29.8 | 8.33 | 22.8 | 1.23 | 6.23 | 6.08 | |
| Rouissat | D3F8 | M | 3545470.7 | 732837.6 | 332.4 | 1996 | 6.16 | 25.3 | 7.22 | 1.71 | 29.81 | 8.33 | 22.86 | 1.23 | 6.23 | 6.08 | |
| Ain El Arch | D3F26 | M | 3534843.9 | 723381.8 | 93.6 | 1996 | 5.11 | 25.1 | 7.45 | 1.56 | 34.68 | 8.94 | 23.98 | 0.87 | 8.38 | 6.5 | |
| St. pompage chott | D5F80 | S | 3541656.9 | 723521.9 | 224.1 | 1996 | 3.69 | 25.4 | 7.67 | 2.28 | 42.22 | 13.53 | 36.77 | 1.12 | 7.43 | 9.73 | |

M = Mio-pliocene aquifer; S = Senonian aquifer.

HESSD

doi:10.5194/hess-2015-385

**Chemical and isotopic data from groundwaters in Sahara**

R. Slimani et al.

Discussion Paper | Discussion Paper | Discussion Paper | Discussion Paper |

HESSD

doi:10.5194/hess-2015-385

**Chemical and isotopic data from groundwaters in Sahara**

R. Slimani et al.

**Table 3.** Field and analytical data for the Phreatic aquifer.

| Locality | Site | Lat. | Long. | Elev. | Date | EC | $t$ | pH | Alk. | Cl⁻ | SO₄²⁻ | Na⁺ | K⁺ | Mg²⁺ | Ca²⁺ | Br⁻ |
|---|---|---|---|---|---|---|---|---|---|---|---|---|---|---|---|---|
| | | | | /m | | mS cm⁻¹ | °C | | | | | mmol L⁻¹ | | | | |
| Khezana | P433 | 3597046 | 719626 | 118 | 20/01/2013 | 2.09 | 22.7 | 9.18 | 1.56 | 12.02 | 7.3 | 13 | 0.99 | 4.34 | 2.8 | |
| Khezana | P433 | 3597046 | 719626 | 118 | 1996 | 2 | 22.1 | 8.86 | 1.46 | 12 | 6.87 | 11.57 | 0.93 | 4.4 | 2.9 | |
| Hassi Miloud | P059 | 3547216 | 718358 | 124 | 27/01/2013 | 2.1 | 23.9 | 8.15 | 1.86 | 13 | 7.3 | 12.6 | 1.25 | 4.43 | 3.43 | 0.024 |
| Ain Kheir | PL06 | | | | 1996 | 4.01 | 23.79 | 7.52 | 1.86 | 14.15 | 17.89 | 15.89 | 0.61 | 10.61 | 7.5 | |
| Hassi Naga | PLX3 | 3584761.4 | 717604.5 | 125 | 20/01/2013 | 2.93 | 23 | 8.09 | 2.04 | 17.7 | 9.4 | 16.6 | 0.93 | 5.75 | 5 | 0.031 |
| | LTP 30 | | | | 1996 | 4.08 | 23.73 | 7.12 | 5.25 | 18.21 | 9.97 | 24.29 | 0.41 | 1.43 | 8.13 | |
| Maison de culture | PL31 | 3537988 | 720114 | 124 | 1996 | 2.51 | 23.83 | 8.08 | 1.46 | 18.91 | 7.8 | 26.05 | 0.62 | 2.13 | 2.99 | |
| El Bour | P006 | 3564272 | 719421 | 161 | 1996 | 2.96 | 23.43 | 7.88 | 1.27 | 18.98 | 7.74 | 12.41 | 2.69 | 5.32 | 5.31 | |
| Hassi Miloud | P059 | 3547216 | 718358 | 124 | 1996 | 2.77 | 23.45 | 7.83 | 2.27 | 20.83 | 9.366 | 34.17 | 4.25 | 1.35 | 0.86 | |
| Oglet Larbaa | P430 | 3567287.5 | 730058.8 | 139 | 24/01/2013 | 4.5 | 27.5 | 8.29 | 3.29 | 22.1 | 12.4 | 21.8 | 2.59 | 8.61 | 5.47 | |
| Maison de culture | PL31 | 3537988 | 720114 | 124 | 28/01/2013 | 3.7 | 22.2 | 8.23 | 4.22 | 22.6 | 8.6 | 28.4 | 2.21 | 4.01 | 3.17 | |
| Frane El Koum | P401 | 3572820.2 | 719721.4 | 112 | 20/01/2013 | 3.44 | 27.5 | 7.52 | 2.21 | 23.3 | 13.4 | 21.8 | 1.86 | 8.25 | 6.28 | 0.032 |
| Gherbouz | PL15 | 3537962 | 718744 | 134 | 1996 | 2.47 | 23.47 | 7.72 | 2.99 | 23.54 | 13.97 | 50.56 | 2.82 | 0.98 | 0.25 | |
| Bour El Haicha | P408 | 3544999.3 | 719930.6 | 110 | 1996 | 2.43 | 23.46 | 7.75 | 2.39 | 24.16 | 13.23 | 41.89 | 6.08 | 2.34 | 0.84 | |
| Station d'épuration | PL30 | 3538398 | 721404 | 130 | 1996 | 5.51 | 23.80 | 7.39 | 3.01 | 24.32 | 21.22 | 24.26 | 0.88 | 20.16 | 2.23 | |
| Frane Ank Djemel | P422 | 3575339 | 718875 | 109 | 20/01/2013 | 4.08 | 24.2 | 8.38 | 4.39 | 25.3 | 9.5 | 23.7 | 1.77 | 4.18 | 7.91 | 0.025 |
| Route Ain Bida | PLX2 | 3537323.9 | 724063.3 | 127 | 1996 | 4.7 | 23.61 | 7.22 | 2.02 | 25.68 | 10.36 | 14.83 | 0.24 | 9.33 | 7.36 | |
| H Chegga | PLX4 | 3577944.8 | 714428.5 | 111 | 20/01/2013 | 4.1 | 25.2 | 7.61 | 3.03 | 26.2 | 9.8 | 24 | 2.32 | 4.96 | 7.46 | 0.033 |
| Hassi Miloud | P058 | 3547329.7 | 716520.7 | 129 | 27/01/2013 | 3.66 | 24.6 | 8.1 | 3 | 27.7 | 10.6 | 19 | 2.29 | 9.09 | 6.55 | 0.033 |
| Route Ain Moussa | P057 | 3548943 | 717353 | 133 | 1996 | 5.3 | 23.44 | 7.69 | 1.34 | 28.21 | 11.48 | 17.58 | 2.03 | 11.48 | 5.8 | |
| Route El Goléa | P115 | 3533586 | 714060 | 141.6 | 1996 | 2.62 | 23.68 | 7.65 | 2.84 | 28.77 | 14.52 | 58.74 | 0.03 | 0.83 | 0.73 | |
| Mekmahad | PL05 | 3537109.4 | 718419.1 | 137 | 1996 | | 23.87 | 7.76 | 1.75 | 30.87 | 16.66 | 24.9 | 0.97 | 15.69 | 4.49 | |
| Polyclinique Belabès | PL18 | 3537270 | 721119 | 119 | 31/01/2013 | 4.67 | 22.2 | 7.89 | 1.78 | 31.2 | 15.4 | 21.3 | 3.87 | 11.17 | 8.37 | |
| H Chegga | PLX4 | 3577944.8 | 714428.5 | 111 | 1996 | 4.49 | 23.67 | 7.58 | 1.5 | 31.52 | 10.08 | 20.05 | 5.87 | 7.53 | 6.5 | |
| Route El Goléa | P116 | 3532463 | 713715 | 117 | 1996 | 5.62 | 23.69 | 7.62 | 1.45 | 31.94 | 12.83 | 22.23 | 0.8 | 10.55 | 7.89 | |
| Gherbouz | PL15 | 3537962 | 718744 | 134 | 21/01/2013 | 4.65 | 23.3 | 8.16 | 1.78 | 32.4 | 14.6 | 27.8 | 0.8 | 6.76 | 10.83 | |
| Route El Goléa | P117 | 3531435 | 713298 | 111 | 1996 | 4.77 | 23.70 | 7.70 | 1.55 | 32.81 | 12.85 | 30.18 | 0.96 | 9.19 | 5.74 | |
| Route Ain Moussa | P057 | 3548943 | 717353 | 133 | 26/01/2013 | 5.7 | 26.2 | 7.64 | 2.48 | 33.5 | 11.9 | 27.7 | 5.93 | 5.98 | 7.57 | |
| Ecole paramédicale | PL32 | 3538478 | 720170 | 131 | 21/01/2013 | 5.72 | 22.9 | 8.21 | 1.96 | 33.6 | 12.1 | 29.2 | 3.35 | 6.36 | 8.17 | |
| DSA | PL10 | 3537055 | 719746 | 114 | 1996 | 6.08 | 23.71 | 7.69 | 1.32 | 35.01 | 13.52 | 8.6 | 1.92 | 19.37 | 7.23 | |
| Route El Goléa | P117 | 3531435 | 713298 | 111 | 03/02/2013 | 5.5 | 25 | 7.72 | 3.25 | 35.4 | 13.8 | 37.1 | 3.04 | 8.45 | 5.68 | |
| Route El Goléa | P116 | 3532463 | 713715 | 117 | 03/02/2013 | 5.8 | 22.5 | 8.04 | 1.66 | 36.3 | 11.6 | 28.5 | 3.21 | 6.75 | 8.37 | |
| Station d'épuration | PL30 | 3538398 | 721404 | 130 | 31/01/2013 | 5.29 | 25.1 | 7.84 | 4.1 | 38.4 | 14.6 | 28.5 | 4.45 | 11.62 | 8.14 | |
| Hassi Debich | P416 | 3581097 | 730922 | 106 | 24/01/2013 | 5.5 | 23.7 | 8.86 | 0.35 | 38.6 | 18 | 22.3 | 0.89 | 4.8 | 21.26 | |
| DSA | PL10 | 3537055 | 719746 | 114 | 28/01/2013 | 5.51 | 24.6 | 8.44 | 2.37 | 38.8 | 16.9 | 36.9 | 1.93 | 9.03 | 9.21 | |
| Hôpital | LTPSN2 | 3538292.9 | 720442.9 | 132 | 27/01/2013 | 6.09 | 25.4 | 7.78 | 1.62 | 39.7 | 11.7 | 36 | 8.43 | 5.11 | 5.97 | |
| PARC SONACOM | PL28 | 3536077 | 719558 | 134 | 21/01/2013 | 6.08 | 24.5 | 8.13 | 1.82 | 39.8 | 11.8 | 30.6 | 5.2 | 7.14 | 8.46 | |
| Bour El Haicha | P408 | 3544999.3 | 719930.6 | 110 | 27/01/2013 | 6.22 | 23.1 | 8.07 | 1.82 | 42 | 19.1 | 27.5 | 13.21 | 13.39 | 8.12 | |
| Route Ain Moussa | P056 | 3549933 | 717022 | 128 | 1996 | 7.62 | 23.65 | 7.93 | 0.56 | 42.14 | 10.72 | 18.87 | 1.86 | 12.63 | 9.32 | |
| Route Ain Moussa | P056 | 3549933 | 717022 | 128 | 26/01/2013 | 5.98 | 24.6 | 7.63 | 2.16 | 42.5 | 17.9 | 32.1 | 8.03 | 12.49 | 8.07 | |
| Ecole Okba B. Nafaa | PL41 | 3538660 | 719831 | 127 | 31/01/2013 | 6.26 | 24.1 | 7.68 | 2.11 | 44.9 | 13.2 | 36.2 | 11.8 | 6.32 | 6.68 | |



**Table 4.** Field and analytical data for the Phreatic aquifer (continued).

| Locality | Site | Lat. | Long. | Elev. | Date | EC | $t$ | pH | Alk. | Cl− | $SO_4^{2-}$ | $Na^+$ | $K^+$ | $Mg^{2+}$ | $Ca^{2+}$ |
|---|---|---|---|---|---|---|---|---|---|---|---|---|---|---|---|
| | | | m | | | $mS\,cm^{-1}$ | °C | | | | | $mmol\,L^{-1}$ | | | |
| PARC HYDRAULIQUE | P419 | 3539494 | 725605 | 132 | 31/01/2013 | 7.03 | 26.4 | 7.84 | 2.05 | 45.1 | 14.4 | 41.4 | 10.78 | 5.95 | 6.91 |
| Parc hydraulique | PL13 | 3536550 | 720200 | 123 | 21/01/2013 | 7.22 | 24.5 | 7.51 | 3.24 | 47.8 | 14.5 | 44.4 | 10.55 | 6.35 | 6.59 |
| Mekhadma | PL25 | 3536230 | 718708 | 129 | 21/01/2013 | 7.64 | 27.1 | 7.94 | 1.78 | 48 | 14.5 | 42.9 | 6.56 | 7.4 | 7.61 |
| Said Otba | P506 | 3535528.1 | 725075.1 | 126 | 04/02/2013 | 8.32 | 24.3 | 8.12 | 1.71 | 52.6 | 14.6 | 42.8 | 10.97 | 7.51 | 7.83 |
| Said Otba | P506 | 3535528.1 | 725075.1 | 126 | 1996 | 6.7 | 23.28 | 7.46 | 1.8 | 54.39 | 17.58 | 33.32 | 4.11 | 22.16 | 5.17 |
| Mekhadma | P566 | 3540433.1 | 719661.3 | 115 | 27/01/2013 | 9 | 24.6 | 7.64 | 1.72 | 62.5 | 15.2 | 71.6 | 3.03 | 4.61 | 6.06 |
| Mekhadma | PL17 | 3536908 | 718511 | 130 | 21/01/2013 | 9.4 | 24.5 | 8.06 | 3.39 | 63.2 | 15.6 | 77.2 | 2.51 | 4.08 | 5.11 |
| Palm. Gara Krima | P413 | 3530116.2 | 722775.1 | 130 | 04/02/2013 | 10.09 | 30.2 | 7.91 | 1.63 | 63.6 | 21.5 | 88.3 | 4.08 | 4.21 | 4.65 |
| Mekhadma | PL25 | 3536230 | 718708 | 129 | 1996 | 9.5 | 23.72 | 7.96 | 0.63 | 75.57 | 10.62 | 10.22 | 2.64 | 32.94 | 9.54 |
| Said Otba (Bab sbaa) | P066 | 3542636.5 | 718957.4 | 126 | 1996 | 7.75 | 23.48 | 7.62 | 1.51 | 80.23 | 12.45 | 45.87 | 2.46 | 23.59 | 5.91 |
| CEM Malek B. Nabi | PL03 | 3540010.9 | 725738.1 | 130 | 1996 | 7.34 | 23.86 | 7.60 | 3.04 | 84.14 | 30.58 | 108.55 | 2.23 | 10.17 | 8.99 |
| ENTV | PL21 | 3536074 | 721268 | 128 | 1996 | 9.73 | 23.82 | 7.25 | 4.46 | 84.26 | 23.68 | 61.62 | 3.75 | 33.53 | 1.88 |
| Hôtel Transat | PL23 | 3538419 | 720950 | 126 | 28/01/2013 | 15 | 24.2 | 8.2 | 4.53 | 86.6 | 16.7 | 79.9 | 3.21 | 14.54 | 6.85 |
| ENTV | PL21 | 3536074 | 721268 | 128 | 28/01/2013 | 16.41 | 25.7 | 7.45 | 1.97 | 99.9 | 17.4 | 85.5 | 5.7 | 15.66 | 7.6 |
| Mekmahad | PL05 | 3537109.4 | 718419.1 | 137 | 21/01/2013 | 16.8 | 24.8 | 7.64 | 2.02 | 101.3 | 17.7 | 85.9 | 5.85 | 16.69 | 7.59 |
| Beni Thour | PL44 | 3536039.3 | 721673.9 | 134 | 1996 | 4.68 | 23.85 | 7.19 | 2.74 | 109.75 | 67.21 | 134.67 | 5.71 | 42.02 | 8.77 |
| Tazegrart | PLSN1 | 3537675 | 719416 | 125 | 22/01/2013 | 17.08 | 24.9 | 8 | 3.41 | 114.2 | 18.1 | 92.9 | 12.8 | 16.85 | 7.81 |
| CEM Malek B. Nabi | PL03 | 3540010.9 | 725738.1 | 130 | 27/01/2013 | 10.84 | 23.1 | 7.54 | 3.29 | 117.3 | 14.7 | 116.4 | 2.06 | 8.99 | 7.24 |
| El Bour | P006 | 3564272 | 719421 | 161 | 03/02/2013 | 18.31 | 23.6 | 7.76 | 6.26 | 131.9 | 18.1 | 96.3 | 8.61 | 27.11 | 7.99 |
| Ain Moussa | P015 | 3551711 | 720591 | 103 | 1996 | 12.42 | 23.62 | 7.71 | 2.38 | 134.68 | 28.2 | 72.98 | 3.1 | 52.44 | 6.25 |
| Station de pompage | PL04 | 3541410.1 | 723501.1 | 138 | 27/01/2013 | 19.01 | 26.4 | 7.85 | 4.03 | 138 | 16.7 | 108.8 | 13.06 | 19.51 | 8.72 |
| Drain Chott Ouargla | D.Ch | | | | 1996 | | 23.88 | 7.67 | 2.68 | 142.22 | 24.5 | 96.31 | 3.16 | 44.22 | 3.02 |
| Beni Thour | PL44 | 3536039.3 | 721673.9 | 134 | 28/01/2013 | 20.18 | 25.8 | 7.8 | 4.96 | 153 | 17.7 | 125.9 | 6.29 | 22.83 | 8.08 |
| CNMC | PL27 | 3535474 | 718407 | 126 | 21/01/2013 | 21.23 | 24.8 | 8.11 | 1.7 | 169.4 | 18.4 | 130.3 | 4.89 | 27.81 | 8.63 |
| Bamendil | P076 | 3540137 | 716721 | 118 | 26/01/2013 | 22.31 | 27.2 | 7.57 | 4.33 | 171.5 | 17.1 | 130.8 | 6.32 | 28.01 | 8.83 |
| N'Goussa | P041 | 3559563 | 716543 | 135 | 26/01/2013 | 25.94 | 24.5 | 8.18 | 7.95 | 208.6 | 13.4 | 198.9 | 3.61 | 11.81 | 8.75 |
| N'Goussa | P009 | 3559388 | 717707 | 123 | 26/01/2013 | 27.51 | 28.4 | 8.39 | 11.45 | 208.8 | 15.8 | 195.1 | 2.65 | 18.7 | 9.01 |
| | LTP16 | | | | 1996 | 11.53 | 23.78 | 7.48 | 3.84 | 213.35 | 48.63 | 147.9 | 7.46 | 75.31 | 4.25 |
| | P100 | | | | 1996 | 17.18 | 23.64 | 7.59 | 3.37 | 235.01 | 46.44 | 264.84 | 4.74 | 25.57 | 5.56 |
| Chott Adjadja Aven | PLX1 | 3540758.8 | 726115.6 | 132 | 28/01/2013 | 32.93 | 23.4 | 7.95 | 4.44 | 245.6 | 20.9 | 141.4 | 26.88 | 44.56 | 17.66 |
| Route Frane | P003 | 3569043 | 721496 | 134 | 02/02/2013 | 31.03 | 23.5 | 8.01 | 6.91 | 252.7 | 17.9 | 208.2 | 9.41 | 29.99 | 10.03 |
| El Bour-N'gouca | P007 | 3562236 | 718651 | 129 | 26/01/2013 | 30.07 | 28.4 | 7.76 | 5.42 | 254.7 | 15.5 | 209.2 | 10.43 | 28.82 | 7.51 |
| Route Ain Bida | PLX2 | 3537323.9 | 724063.2 | 127 | 21/01/2013 | 43.25 | 25.7 | 8.07 | 5.15 | 262.2 | 93 | 270.4 | 15.5 | 62.77 | 21.46 |
| Ain Moussa | P015 | 3551711 | 720591 | 103 | 25/01/2013 | 32.02 | 22.7 | 8.03 | 2.95 | 263 | 15.4 | 206.9 | 6.56 | 32.12 | 9.95 |
| Ain Moussa | P402 | 3549503 | 721514 | 138 | 25/01/2013 | 60 | 28.7 | 8.6 | 7.69 | 313.2 | 93.9 | 442.8 | 23.26 | 12.56 | 10.17 |
| Route Frane | P001 | 3572148 | 722366 | 127 | 1996 | | 23.63 | 8.37 | 4 | 323.62 | 58.13 | 331.43 | 5.01 | 49.77 | 3.97 |
| Ain Moussa | P014 | 3551466 | 719339 | 131 | 1996 | | 23.40 | 7.31 | 3.98 | 336.96 | 64.29 | 328.67 | 5.53 | 62.37 | 5.45 |
| N'Goussa | P019 | 3562960 | 717719 | 113 | 02/02/2013 | 60.58 | 27.8 | 7.65 | 6.02 | 356.2 | 96 | 432.5 | 29.77 | 21.02 | 26.23 |
| N'Goussa | P018 | 3562122 | 716590 | 110 | 26/01/2013 | 61.06 | 26.2 | 8.42 | 6.46 | 372.4 | 82.3 | 347.1 | 22.64 | 60.71 | 26.63 |
| Ain Moussa | P014 | 3551466 | 719339 | 131 | 25/01/2013 | 49.04 | 25.2 | 7.89 | 1.8 | 399.7 | 21.1 | 389.3 | 2.41 | 18.97 | 7.39 |
| Route Sedrata | P113 | 3535586 | 714576 | 105 | 03/02/2013 | 62.24 | 24.8 | 8.2 | 5.96 | 414.8 | 83.8 | 362.7 | 33.34 | 70.23 | 26.51 |
| N'Goussa | P009 | 3559388 | 717707 | 123 | 1996 | | 23.27 | 7.84 | 2.4 | 426.85 | 57.81 | 393.83 | 9.13 | 59.13 | 12.02 |

HESSD

doi:10.5194/hess-2015-385

Chemical and isotopic data from groundwaters in Sahara

R. Slimani et al.

Discussion Paper | Discussion Paper | Discussion Paper | Discussion Paper

**HESSD**

doi:10.5194/hess-2015-385

**Chemical and isotopic data from groundwaters in Sahara**

R. Slimani et al.

**Table 5.** Field and analytical data for the Phreatic aquifer (continued).

| Locality | Site | Lat. | Long. | Elev. | Date | EC | $t$ | pH | Alk. | Cl– | $SO_4^{2-}$ | $Na^+$ | $K^+$ | $Mg^{2+}$ | $Ca^{2+}$ |
|---|---|---|---|---|---|---|---|---|---|---|---|---|---|---|---|
| | | | | m | | mS cm⁻¹ | °C | | | | | mmol L⁻¹ | | | |
| Route Frane | P001 | 3572148 | 722366 | 127 | 02/02/2013 | 66.16 | 28.3 | 7.24 | 6.49 | 468.7 | 101.5 | 350.3 | 25.96 | 116.21 | 35.31 |
| Sebkhet Safioune | P031 | 3577804 | 720172 | 120 | 1996 | | 23.75 | 7.31 | 6.32 | 481.83 | 43.35 | 326.82 | 12.61 | 94.15 | 23.56 |
| Sebkhet Safioune | P031 | 3577804 | 720172 | 120 | 02/02/2013 | 75.96 | 27.9 | 8.06 | 5.85 | 500.3 | 110.3 | 470.5 | 28.67 | 79.12 | 35.47 |
| Route Frane | P002 | 3570523 | 722028 | 108 | 1996 | | 23.81 | 7.76 | 6.29 | 522.39 | 182.95 | 653.78 | 9.97 | 104.7 | 10.99 |
| Sebkhet Safioune | P030 | 3577253 | 721936 | 130 | 1996 | | 23.52 | 7.72 | 4.43 | 527.7 | 123.48 | 533.79 | 11.59 | 106.21 | 10.65 |
| Oum Raneb | P012 | 3554089 | 718612 | 114 | 25/01/2013 | 64.05 | 30.3 | 7.83 | 7.77 | 534.3 | 20.9 | 529.6 | 6.41 | 19.73 | 4.73 |
| Oum Raneb | P012 | 3554089 | 718612 | 114 | 1996 | | 23.41 | 7.46 | 2.72 | 539.35 | 60.64 | 413.55 | 5.55 | 112.77 | 9.42 |
| ANK Djemel | P423 | 3540881 | 723178 | 102 | 31/01/2013 | 90.8 | 23.5 | 7.48 | 6.19 | 636.5 | 101.3 | 495.5 | 38.31 | 125.81 | 30.32 |
| Said Otba-Chott | P096 | 3540265 | 724729 | 111 | 1996 | | 23.59 | 7.71 | 3.69 | 645.07 | 78.46 | 357.28 | 5.89 | 208.4 | 12.86 |
| Sebkhet Safioune | P030 | 3577253 | 721936 | 130 | 03/02/2013 | 64.66 | 23.1 | 7.83 | 3.71 | 671.8 | 90.3 | 742.9 | 15.97 | 41.46 | 7.65 |
| N'Goussa | P017 | 3560256 | 715781 | 130 | 26/01/2013 | 100.1 | 31 | 7.13 | 3.78 | 679.3 | 114.1 | 597.8 | 10.71 | 125.85 | 26.29 |
| ANK Djemel | P021 | 3573943 | 723161 | 105 | 1996 | | 23.55 | 7.43 | 4.24 | 700.77 | 154.45 | 605.68 | 53.6 | 163.08 | 14.24 |
| Station de pompage | PL04 | 3541410.1 | 723501.1 | 138 | 1996 | | 23.57 | 7.42 | 2.37 | 716.27 | 34.75 | 560.07 | 7.04 | 99.58 | 11.04 |
| Route Frane | P002 | 3570523 | 722028 | 108 | 02/02/2013 | 62.82 | 26.9 | 7.57 | 1.65 | 748.5 | 62.6 | 651.5 | 14.72 | 77.72 | 27.29 |
| Said Otba-Chott | P096 | 3540265 | 724729 | 111 | 03/02/2013 | 68.31 | 25.9 | 8.7 | 1.24 | 771 | 53.1 | 615.9 | 23.46 | 69.64 | 50.39 |
| N'Goussa | P019 | 3562960 | 717719 | 113 | 1996 | | 23.30 | 7.72 | 2.42 | 779.13 | 77.13 | 711.46 | 9.23 | 95.59 | 12.05 |
| Said Otba(Bab sbaa) | P066 | 3542636.5 | 718957.4 | 126 | 03/02/2013 | 150.6 | 26.2 | 7.18 | 12.29 | 799.1 | 283 | 1249.7 | 18.95 | 37.63 | 18.06 |
| ANK Djemel | P021 | 3573943 | 723161 | 105 | 24/01/2013 | 82.28 | 29.6 | 7.64 | 2.35 | 800.4 | 94.4 | 824 | 10.99 | 53.35 | 25.39 |
| N'Goussa | P018 | 3562122 | 716590 | 110 | 1996 | | 23.29 | 7.46 | 1.24 | 818.67 | 81 | 244.21 | 49.54 | 319.35 | 24.76 |
| Oum Raneb | P162 | 3546133 | 725129 | 98 | 25/01/2013 | 160 | 30.7 | 7.15 | 2.43 | 842.8 | 289.9 | 1309.9 | 13.3 | 33.47 | 17.74 |
| Route Sedrata | P113 | 3535586 | 714576 | 105 | 1996 | | 23.66 | 7.70 | 2.81 | 954.89 | 124.85 | 997.52 | 13.3 | 86.69 | 11.67 |
| Oum Raneb | PZ12 | 3547234 | 722931 | 110 | 05/02/2013 | 114.9 | 27.4 | 7.44 | 2.88 | 980.1 | 15.5 | 930.8 | 7.53 | 23.9 | 14.24 |
| Hôtel Transat | PL23 | 3538419 | 720950 | 126 | 1996 | | 23.49 | 7.37 | 3 | 1103.31 | 94.49 | 707.81 | 19.14 | 270.91 | 13.3 |
| Sebkhet Safioune | P023 | 3577198 | 725726 | 99 | 1996 | | 23.32 | 7.42 | 2.25 | 1176.99 | 91.14 | 1058.21 | 11.72 | 133.47 | 12.41 |
| Sebkhet Safioune | P034 | 3579698 | 725633 | 97 | 05/02/2013 | 130 | 34.9 | 8.08 | 1.76 | 1189.1 | 14.7 | 1055.1 | 18.27 | 56.37 | 17.38 |
| Sebkhet Safioune | P023 | 3577198 | 725726 | 99 | 05/02/2013 | 117.9 | 29.4 | 8.19 | 1.85 | 1209.3 | 15.6 | 1129.4 | 8.38 | 42.85 | 10.15 |
| Chott Adjadja | PLX1 | 3540758.8 | 726115.6 | 132 | 1996 | | 23.60 | 8.02 | 3.82 | 1296.65 | 134.01 | 1458.73 | 5.24 | 47.98 | 4.34 |
| Sebkhet Safioune | P063 | 3545586.8 | 725667.4 | 99 | 1996 | | 23.50 | 7.46 | 1.94 | 1379.35 | 139.61 | 1257.42 | 18.6 | 182.26 | 10.03 |
| | LTP06 | | | | 1996 | | 23.77 | 7.64 | 7.84 | 1638.66 | 712.09 | 2621.61 | 41.55 | 190.51 | 13.34 |
| Bamendil | P076 | 3540137 | 716721 | 118 | 1996 | | 23.53 | 7.71 | 5.72 | 1743.55 | 143.36 | 1321.87 | 26.85 | 331.38 | 12.26 |
| El Bour-N'gouca | P007 | 3562236 | 718651 | 129 | 1996 | | 23.26 | 7.67 | 1.41 | 1860.53 | 91.55 | 1434.73 | 26.2 | 278.77 | 13.25 |
| Sebkhet Safioune | P063 | 3545586.8 | 725667.4 | 99 | 05/02/2013 | 178.9 | 26.7 | 7.67 | 1.43 | 1887.9 | 92.9 | 1455.8 | 26.66 | 282.88 | 13.44 |
| | P044 | | | | 1996 | | 23.39 | 7.79 | 4.53 | 2106.07 | 18.27 | 1765.47 | 27.33 | 171.23 | 6.54 |
| | P093 | | | | 1996 | | 23.58 | 7.49 | 1.49 | 2198.58 | 182.08 | 1957.53 | 29.49 | 278.18 | 10.44 |
| | P042 | | | | 1996 | | 23.42 | 7.59 | 1.1 | 2330.85 | 101.22 | 1963.71 | 52.19 | 248.1 | 11.24 |
| | P068 | | | | 1996 | | 23.51 | 7.54 | 3.35 | 2335.67 | 222.08 | 2302.25 | 26.84 | 219.9 | 7.19 |
| Oum Raneb | PZ12 | 3547234 | 722931 | 110 | 1996 | | 23.31 | 7.59 | 2.21 | 2405.55 | 109.92 | 2178.55 | 25.23 | 199.35 | 12.65 |
| Hassi Debich | P416 | 3581097 | 730922 | 106 | 1996 | | 23.33 | 7.84 | 4.33 | 2433.73 | 178.87 | 2361.09 | 24.34 | 196.07 | 9.2 |
| N'Goussa | P041 | 3559563 | 716543 | 135 | 1996 | | 23.38 | 7.94 | 2.13 | 2599.74 | 324.58 | 2878.99 | 44.57 | 152.83 | 10.97 |
| Sebkhet Safioune | P034 | 3579698 | 725633 | 97 | 1996 | | 23.34 | 7.85 | 1.95 | 2752 | 134.14 | 2616.77 | 24.42 | 180.14 | 10.48 |
| | P039 | | | | 1996 | | 23.37 | 6.87 | 1.94 | 4189.51 | 201.44 | 4042.62 | 17.9 | 257.81 | 9.23 |
| Sebkhet Safioune | P074 | | | | 1996 | | 23.54 | 6.47 | 4.17 | 4356.48 | 180.88 | 2759.9 | 57.4 | 930.06 | 22.63 |
| Sebkhet Safioune | P037 | | | | 1996 | | 23.36 | 6.92 | 1.52 | 4953.84 | 184.54 | 4611.06 | 2.9 | 347.57 | 7.86 |
| Sebkhet Safioune | P036 | | | | 1996 | | 23.35 | 7.54 | 1.4 | 4972.75 | 108.12 | 4692.23 | 36.84 | 221.13 | 9.63 |

## HESSD

doi:10.5194/hess-2015-385

**Chemical and isotopic data from groundwaters in Sahara**

R. Slimani et al.

**Table 6.** Isotopic data $^{18}$O and $^{3}$H and chloride concentration in Continental Intercalaire, Complexe Terminal and Phreatic aquifers.

| Piezometer | Cl⁻ mmol L⁻¹ | δ¹⁸O ‰ | ³H UT | Piezometer | Cl⁻ mmol L⁻¹ | δ¹⁸O ‰ | ³H UT | Piezometer | Cl⁻ mmol L⁻¹ | δ¹⁸O ‰ | ³H UT |
|---|---|---|---|---|---|---|---|---|---|---|---|
| \multicolumn Phreatic aquifer ||||||||||||
| P007 | 1860.5 | −2.49 | 0 | PL15 | 23.54 | −7.85 | 0.6(1) | P074 | 4356.4 | 3.42 | 6.8(8) |
| P009 | 426.85 | −6.6 | 1.2(3) | P066 | 80.23 | −8.14 | 0.8(1) | PL06 | 14.15 | −8.13 | 1.0(2) |
| P506 | 54.39 | −6.83 | 1.6(3) | PL23 | 1103.32 | −6.1 | 0 | PL30 | 24.32 | −7.48 | 2.4(4) |
| P018 | 818.67 | −2.95 | 6.2(11) | P063 | 1379.3 | −3.4 | 8.7(15) | P002 | 522.39 | −5.71 | 0.6(1) |
| P019 | 779.13 | −4.67 | 5.6(9) | P068 | 2335.6 | −3.04 | 8.8(14) | PL21 | 84.26 | −7.65 | 1.2(2) |
| PZ12 | 2405.5 | −2.31 | 8.1(13) | P030 | 527.7 | −6.57 | 2.4(4) | PL31 | 18.91 | −7.38 | 1.6(3) |
| P023 | 1176.9 | −2.62 | 0.2(1) | P076 | 1743.5 | −5.56 | 2.8(5) | P433 | 12 | −8.84 | 0 |
| P416 | 2433.7 | −7.88 | 5.9(9) | P021 | 700.7 | −5.16 | 2.6(4) | PL03 | 84.14 | −7.35 | 1.7(3) |
| P034 | 2752 | −1.77 | 5.7(9) | PL04 | 716.27 | −2.89 | | PL44 | 109.75 | −8.82 | 1.0(2) |
| P036 | 4972.7 | 3.33 | 2.1(4) | P093 | 2198.5 | −2.64 | 5.1(8) | PL05 | 30.87 | −7.44 | 1.9(3) |
| P037 | 4953.8 | 3.12 | 1.8(3) | P096 | 645.07 | −6.13 | 4.8(8) | P408 | 24.16 | −7.92 | 0 |
| P039 | 4189.5 | 0.97 | 2.2(4) | PLX1 | 1296.6 | −5.6 | 1.1(2) | P116 | 31.94 | −7.18 | 1.1(2) |
| P041 | 2599.7 | −0.58 | 7.3(13) | PLX2 | 25.68 | −7.6 | 1.3(2) | LTP 16 | 213.35 | −7.48 | 1.6(3) |
| P044 | 2106.1 | −4.46 | 2.7(5) | P015 | 134.68 | −6.77 | 3.0(5) | P117 | 32.81 | −6.92 | 0.1 |
| P014 | 336.96 | −6.9 | 2.8(5) | P001 | 323.62 | −4.66 | 2.5(4) | PL10 | 35.01 | −7.31 | 0.2(1) |
| P012 | 539.3 | −6.41 | 2.2(4) | P100 | 235.01 | −5.81 | 0 | PL25 | 75.57 | −7.41 | 0.9(2) |
| P042 | 2330.8 | 2.05 | 6.0(10) | P056 | 42.14 | −7.03 | 2.9(5) | LTP30 | 18.21 | −7.5 | 1.1(2) |
| P006 | 18.98 | −6.64 | 0.5(1) | P113 | 954.89 | −4.75 | 0.8(2) | LTP06 | 1638.6 | −1.97 | 2.8(5) |
| P057 | 28.21 | −7.33 | 1.1(2) | PLX4 | 31.52 | −7.1 | 0.3(1) | P031 | 481.83 | −6.06 | 3.0(5) |
| P059 | 20.83 | −7.81 | 0 | P115 | 28.77 | −2.54 | 6.8(12) | | | | |

| Borehole | Cl⁻ mmol L⁻¹ | δ¹⁸O ‰ | ³H UT | Borehole | Cl⁻ mmol L⁻¹ | δ¹⁸O ‰ | ³H UT | Borehole | Cl⁻ mmol L⁻¹ | δ¹⁸O ‰ | ³H UT |
|---|---|---|---|---|---|---|---|---|---|---|---|
| \multicolumn Complexe Terminal aquifer ||||||||||||
| D5F80 | 42.22 | −7.85 | | D1F138 | 28.92 | −8.13 | 0.7(1) | D2F71 | 13.53 | −8.23 | 0.6(1) |
| D3F8 | 29.81 | −8.14 | 1.4(2) | D3F18 | 21.66 | −8.23 | 0.2(1) | D7F4 | 10.6 | −8.27 | 0.1(1) |
| D3F26 | 34.68 | −7.97 | 0.8(1) | D3F10 | 14.27 | −7.88 | 1.5(2) | D2F66 | 11.02 | −8.3 | |
| D4F94 | 20.05 | −8.18 | 0.6(1) | D6F51 | 28.39 | −7.9 | 0.7(1) | D1F151 | 10.75 | −8.32 | 0.4(1) |
| D6F67 | 18.79 | −8.23 | 3.7(6) | D1F135 | 18.08 | −7.97 | 1.1(2) | D6F64 | 11.36 | −8.28 | 4.3(7) |

| Borehole | Cl⁻ mmol L⁻¹ | δ¹⁸O ‰ | ³H UT | Borehole | Cl⁻ mmol L⁻¹ | δ¹⁸O ‰ | ³H UT | Borehole | Cl⁻ mmol L⁻¹ | δ¹⁸O ‰ | ³H UT |
|---|---|---|---|---|---|---|---|---|---|---|---|
| \multicolumn Continental Intercalaire aquifer ||||||||||||
| Hadeb I | 5.8 | −8.02 | 0 | Hadeb II | 6.19 | −7.93 | 0.1(1) | Aouinet Moussa | 6.49 | −7.88 | 1.1(2) |

# HESSD

doi:10.5194/hess-2015-385

**Chemical and isotopic data from groundwaters in Sahara**

R. Slimani et al.

**Table 7.** Statistical parameters for Continental Intercalaire (CI), Complexe Terminal (CT) and Phreatic (Phr) aquifers samples selected on the basis of $\delta^{18}O$ and $Cl^-$ data (see text).

| Aquifer | Size | Parameter | EC mS cm$^{-1}$ | $t$ °C | pH | Alk. | $Cl^-$ | $SO_4^{2-}$ | $Na^+$ | $K^+$ | $Mg^{2+}$ | $Ca^{2+}$ |
|---|---|---|---|---|---|---|---|---|---|---|---|---|
| | | | | | | | | | mmol L$^{-1}$ | | | |
| CI | 11 | Average | 2.2 | 49. | 7.5 | 2.3 | 11. | 4.7 | 10.3 | 0.51 | 3.6 | 2.4 |
| CI | 11 | Stdd. dev. | 0.3 | 2. | 0.2 | 1. | 4.6 | 2.5 | 4.6 | 0.23 | 2. | 1.8 |
| CT | 50 | Average | 3.2 | 23. | 7.8 | 2.3 | 20. | 8.9 | 17. | 1.0 | 5.5 | 5.6 |
| CT | 50 | Stdd. dev. | 1.1 | 2.4 | 0.4 | 0.8 | 7. | 2.6 | 6. | 0.8 | 2.2 | 1.7 |
| Phr pole I | 30 | Average | 3.9 | 24. | 7.9 | 2.3 | 24.7 | 11.8 | 24.2 | 2.1 | 7.2 | 5.3 |
| Phr pole I | 30 | Stdd. dev. | 1.3 | 1.3 | 0.4 | 1. | 6.9 | 3.4 | 11.0 | 1.7 | 5. | 2.7 |
| Phr pole II | 3 | Average | | 23.4 | 7. | 2.4 | 4761. | 158. | 4021. | 32.4 | 500. | 13. |
| Phr pole II | 3 | Stdd. dev. | | 0.1 | 0.5 | 1.6 | 350. | 43. | 1093. | 28. | 378. | 8. |

Discussion Paper | Discussion Paper | Discussion Paper | Discussion Paper

**HESSD**

doi:10.5194/hess-2015-385

**Chemical and isotopic data from groundwaters in Sahara**

R. Slimani et al.

**Table 8.** Summary of mass transfer for geochemical inverse modeling. Phases and thermodynamic database are from Phreeqc 3.0 (Parkhurst and Appelo, 2013).

| Phases | Stoichiometry | CI/CT | CT/Phr I | Rainwater/P036 | PhrI/PhrII 60 %/40 % |
|---|---|---|---|---|---|
| Calcite | $CaCO_3$ | – | −6.62E-06 | −1.88E-01 | −2.26E-01 |
| $CO_2$(g) | $CO_2$ | −6.88E-05 | – | 8.42E-04 | 5.77E-04 |
| Gypsum | $CaSO_4.2H_2O$ | 4.33E-03 | – | 1.55E-01 | 1.67E-01 |
| Halite | NaCl | 7.05E-03 | 3.76E-03 | 6.72E+00 | 1.28E+00 |
| Sylvite | KCl | 2.18E-03 | 1.08E-03 | 4.02E-01 | – |
| Bloedite | $Na_2Mg(SO_4)2.4H_2O$ | – | 1.44E-03 | – | – |
| Huntite | $CaMg_3(CO_3)4$ | – | – | 4.74E-02 | 5.65E-02 |
| Ca ion exchange | $CaX_2$ | −1.11E-03 | – | – | – |
| Mg ion exchange | $MgX_2$ | 1.96E-03 | – | 1.75E-01 | −2.02E-01 |
| Na ion exchange | NaX | – | – | – | 3.92E-01 |
| K ion exchange | KX | −1.69E-03 | – | −3.49E-01 | 1.20E-02 |

Values are in $mol\,kg^{-1}$ $H_2O$. Positive (mass entering solution) and negative (mass leaving solution) phase mole transfers indicate dissolution and precipitation, respectively; – indicates no mass transfer.



Discussion Paper | Discussion Paper | Discussion Paper | Discussion Paper | Discussion Paper |

**HESSD**

doi:10.5194/hess-2015-385

**Chemical and isotopic data from groundwaters in Sahara**

R. Slimani et al.

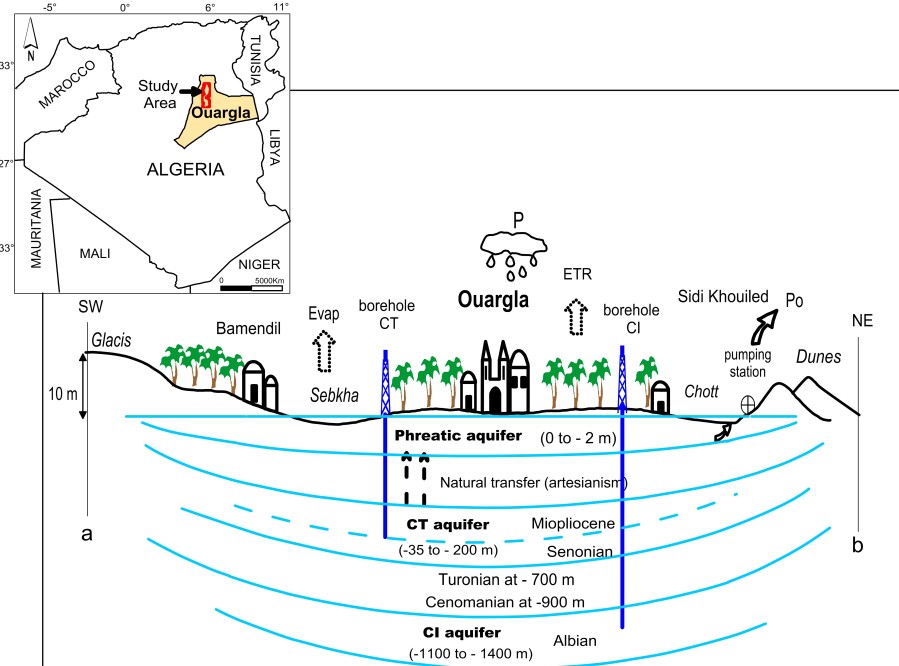

**Figure 1.** Localisation and schematic relations of aquifers in Ouargla. Blue lines represent limits between aquifers, and the names of aquifers are given in bold letters; as the limit between Senonian and Miopliocene aquifers is not well defined, a dashed blue line is used. Names of villages and cities are given in roman (Bamendil, Ouargla, Sidi Khouiled), while geological/geomorphological features are in italic (Glacis, Sebkha, Chott, Dunes). Depths are relative to the ground surface. Letters a and b refer to the cross section (Fig. 2) and to the localisation map (Fig. 3).



# HESSD

doi:10.5194/hess-2015-385

**Chemical and isotopic data from groundwaters in Sahara**

R. Slimani et al.

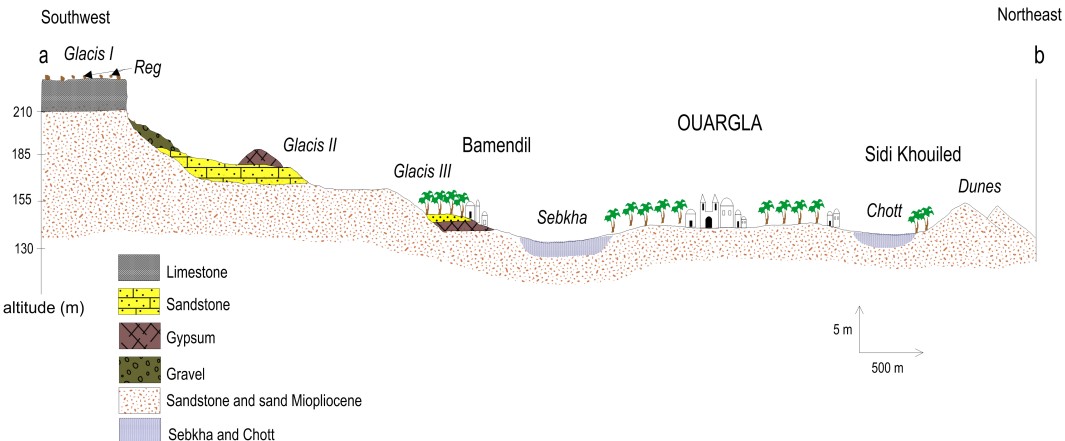

**Figure 2.** Geologic cross section in the region of Ouargla. The blue pattern used for Chott and Sebkha correspond to the limit of the saturated zone.

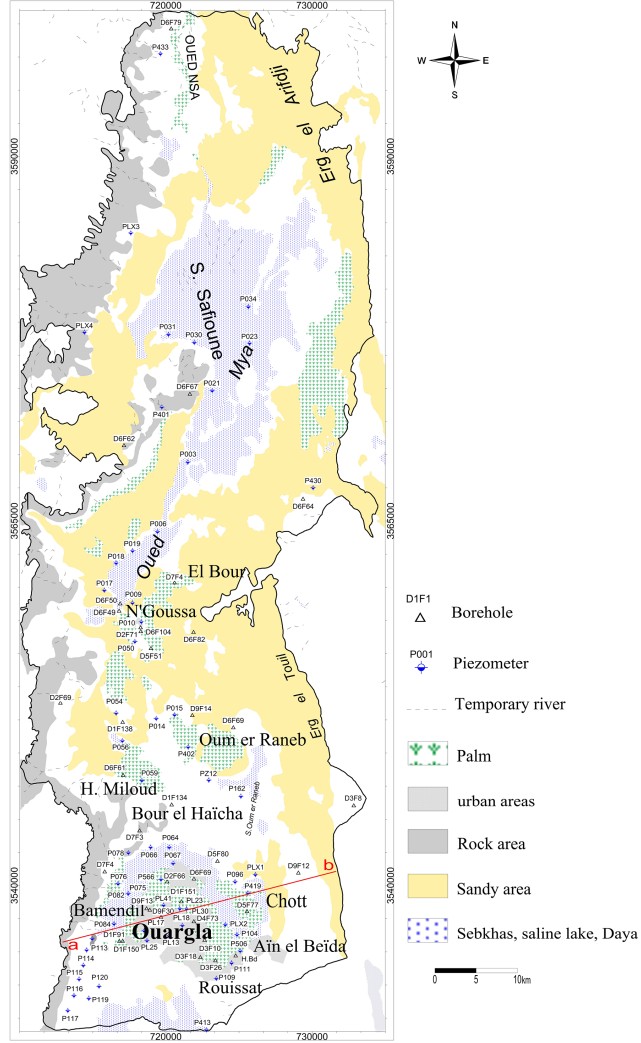

**HESSD**

doi:10.5194/hess-2015-385

**Chemical and isotopic data from groundwaters in Sahara**

R. Slimani et al.

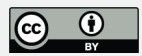

**Figure 3.** Localisation map of sampling points.

**HESSD**

doi:10.5194/hess-2015-385

**Chemical and isotopic data from groundwaters in Sahara**

R. Slimani et al.

**HESSD**

doi:10.5194/hess-2015-385

**Chemical and isotopic data from groundwaters in Sahara**

R. Slimani et al.

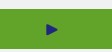

**Figure 4.** Piper diagram for Continental Intercalaire (filled squares), Complexe Terminal (open circles) and Phreatic aquifer (open triangles).



**Figure 5.** Contour maps of the salinity (expressed as global mineralization) in the aquifer system, **(a)** Phreatic aquifer; **(b)** and **(c)** Complexe Terminal [**(b)** Mio-pliocene and **(c)** Senonian]; figures are isovalues of global mineralization (values in g L$^{-1}$).

**HESSD**

doi:10.5194/hess-2015-385

**Chemical and isotopic data from groundwaters in Sahara**

R. Slimani et al.

Discussion Paper | Discussion Paper | Discussion Paper | Discussion Paper |

# HESSD

doi:10.5194/hess-2015-385

## Chemical and isotopic data from groundwaters in Sahara

R. Slimani et al.

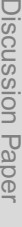

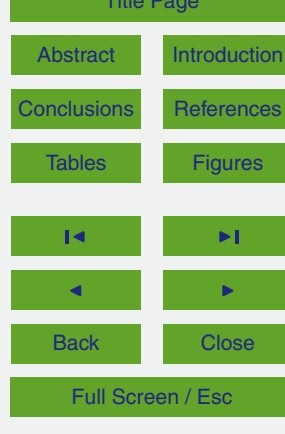

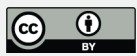

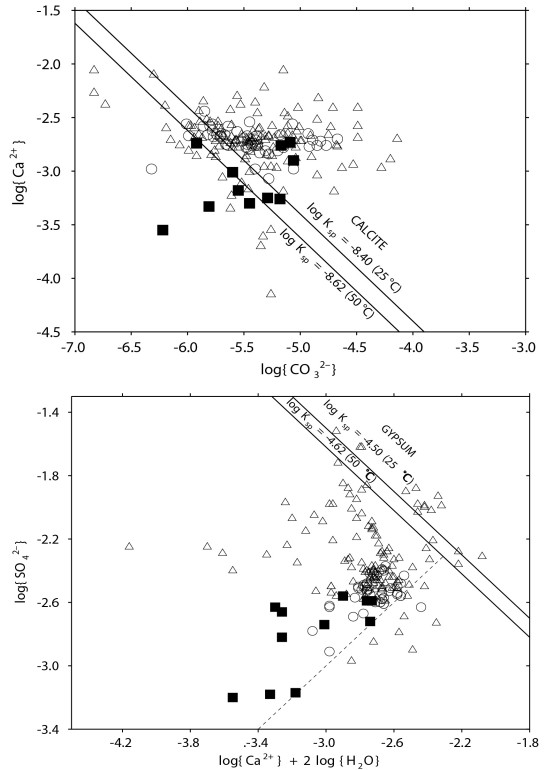

**Figure 6.** Equilibrium diagrams of calcite (top) and gypsum (bottom) for Continental Inter-calaire (filled squares), Complexe Terminal (open circles) and Phreatic aquifer (open triangles). Equilibrium lines are defined as: $\log\{Ca^{2+}\} + \log\{CO_3^{2-}\} = \log K_{sp}$ for calcite, and $\log\{Ca^{2+}\} + 2\log\{H_2O\} + \log\{SO_4^{2-}\} = \log K_{sp}$ for gypsum.

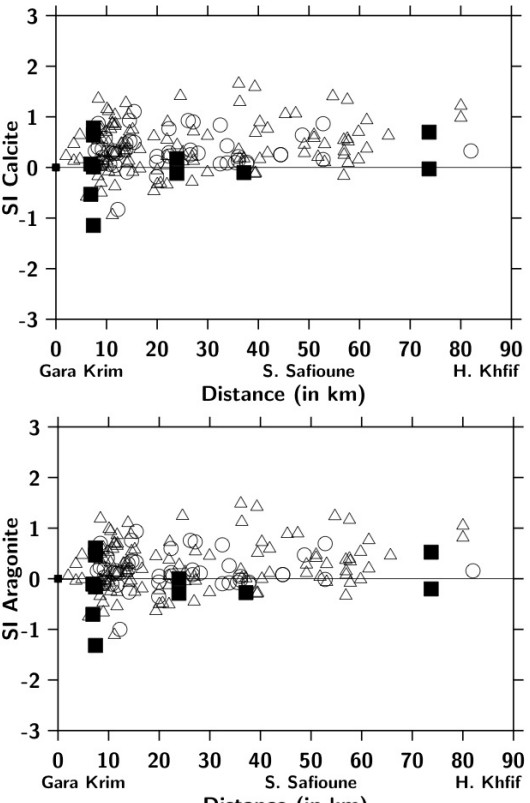

**Figure 7.** Variation of saturation indices of calcite and aragonite with distance from south to north in the region of Ouargla.

**HESSD**

doi:10.5194/hess-2015-385

**Chemical and isotopic data from groundwaters in Sahara**

R. Slimani et al.

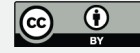

**HESSD**

doi:10.5194/hess-2015-385

**Chemical and isotopic data from groundwaters in Sahara**

R. Slimani et al.

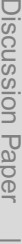

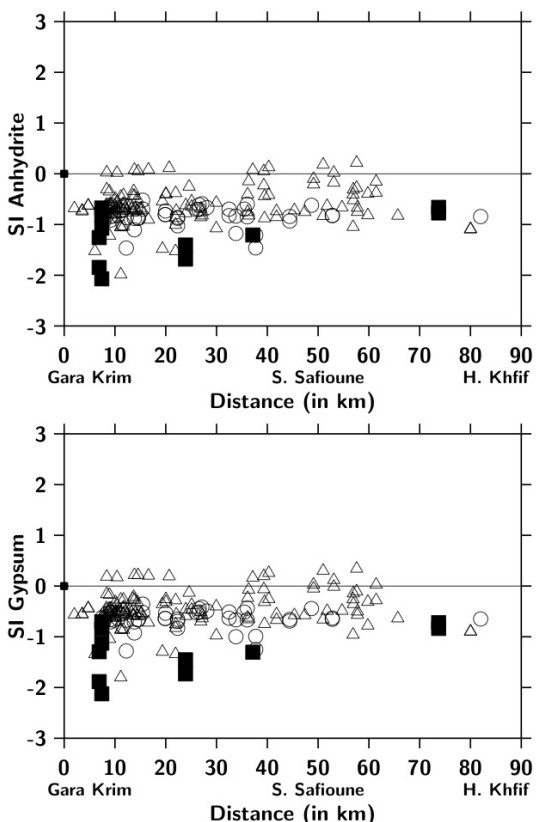

**Figure 8.** Variation of saturation indices of anhydrite and gypsum with distance from south to north in the region of Ouargla.

Interactive Discussion

Discussion Paper | Discussion Paper | Discussion Paper | Discussion Paper

Discussion Paper | Discussion Paper | Discussion Paper | Discussion Paper

**HESSD**

doi:10.5194/hess-2015-385

**Chemical and isotopic data from groundwaters in Sahara**

R. Slimani et al.

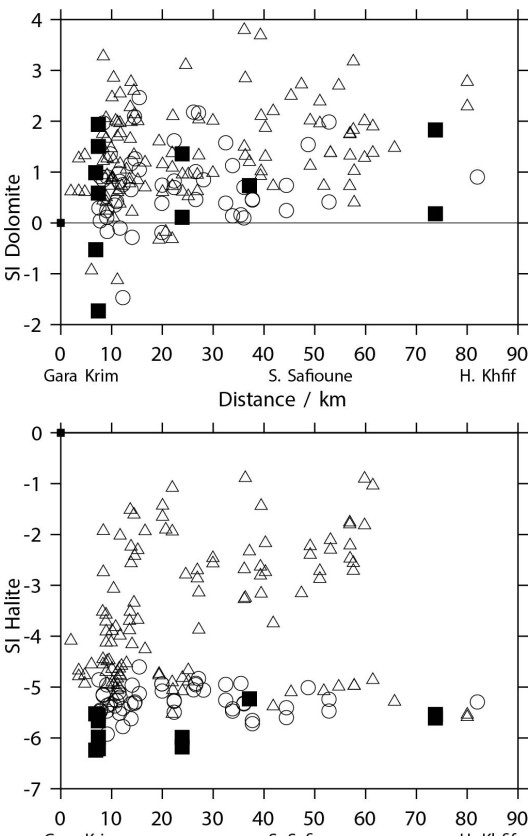

**Figure 9.** Variation of saturation indices of dolomite and halite with distance from south to north in the region of Ouargla.

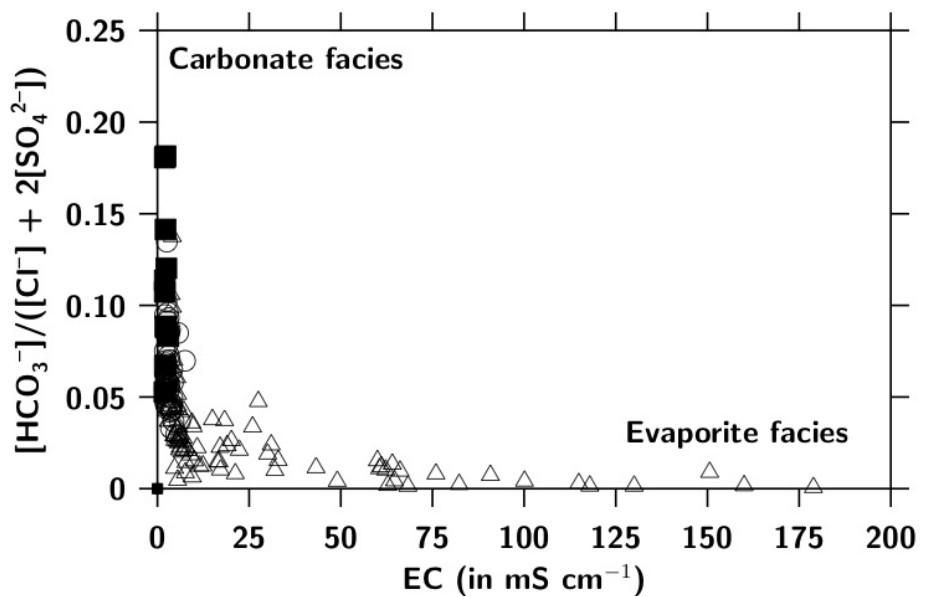

**Figure 10.** Change from carbonate facies to evaporite from Continental Intercalaire (filled squares), Complexe Terminal (open circles) and Phreatic aquifer (open triangles).

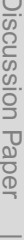

**HESSD**

doi:10.5194/hess-2015-385

**Chemical and isotopic data from groundwaters in Sahara**

R. Slimani et al.

Interactive Discussion

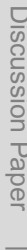

**HESSD**

doi:10.5194/hess-2015-385

**Chemical and isotopic data from groundwaters in Sahara**

R. Slimani et al.

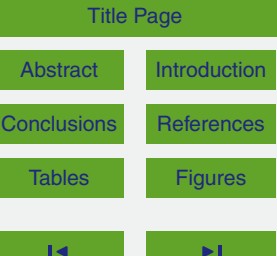

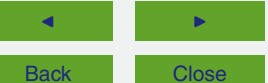



**Figure 11.** Change from sulfate facies to chloride from Continental Intercalaire (filled squares), Complexe Terminal (open circles) and Phreatic aquifer (open triangles).

**HESSD**

doi:10.5194/hess-2015-385

**Chemical and isotopic data from groundwaters in Sahara**

R. Slimani et al.

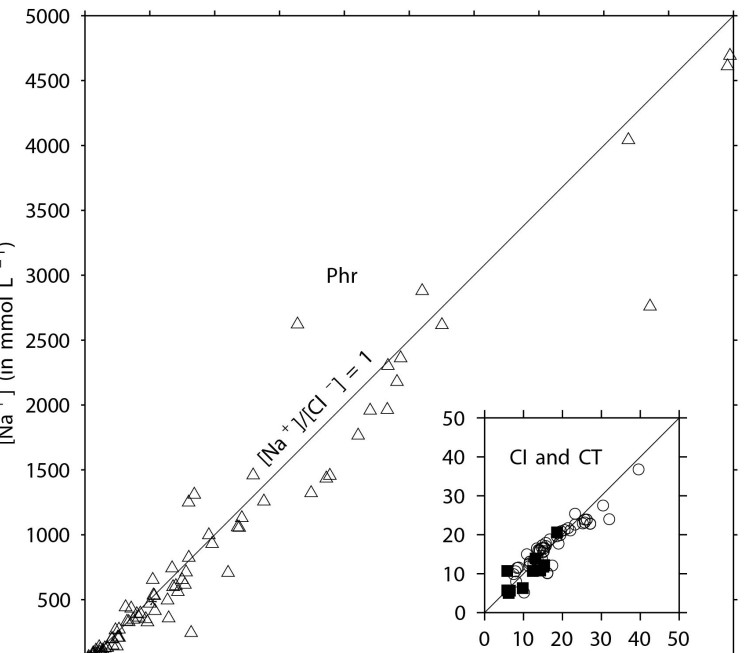

**Figure 12.** Correlation between $Na^+$ and $Cl^-$ concentrations in Continental Intercalaire (filled squares), Complexe Terminal (open circles) and Phreatic aquifer (open triangles). Seawater composition (star) is $[Na^+] = 459.3\,mmol\,L^{-1}$ and $[Cl^-] = 535.3\,mmol\,L^{-1}$ (Stumm and Morgan, 1999, p.899).

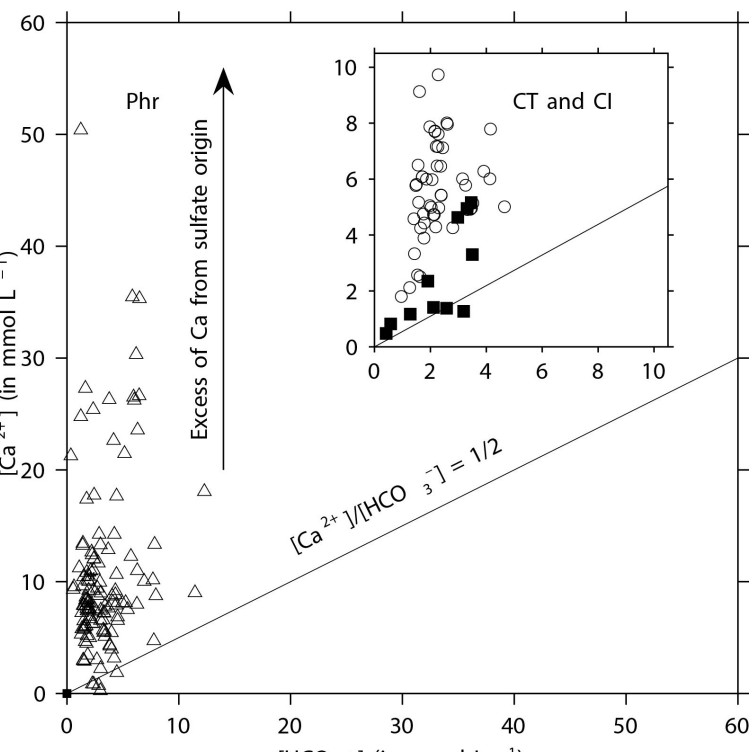

**Figure 13.** Calcium vs. $HCO_3^-$ diagram in Continental Intercalaire (filled squares), Complexe Terminal (open circles), Phreatic aquifer (open triangles) and Seawater composition (star) is $[Ca^{2+}] = 10.2\,\text{mmol}\,L^{-1}$ and $[HCO_3^-] = 2.38\,\text{mmol}\,L^{-1}$ (Stumm and Morgan, 1999, p.899).

**HESSD**

doi:10.5194/hess-2015-385

**Chemical and isotopic data from groundwaters in Sahara**

R. Slimani et al.

Discussion Paper | Discussion Paper | Discussion Paper | Discussion Paper

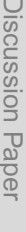

# HESSD

doi:10.5194/hess-2015-385

**Chemical and isotopic data from groundwaters in Sahara**

R. Slimani et al.

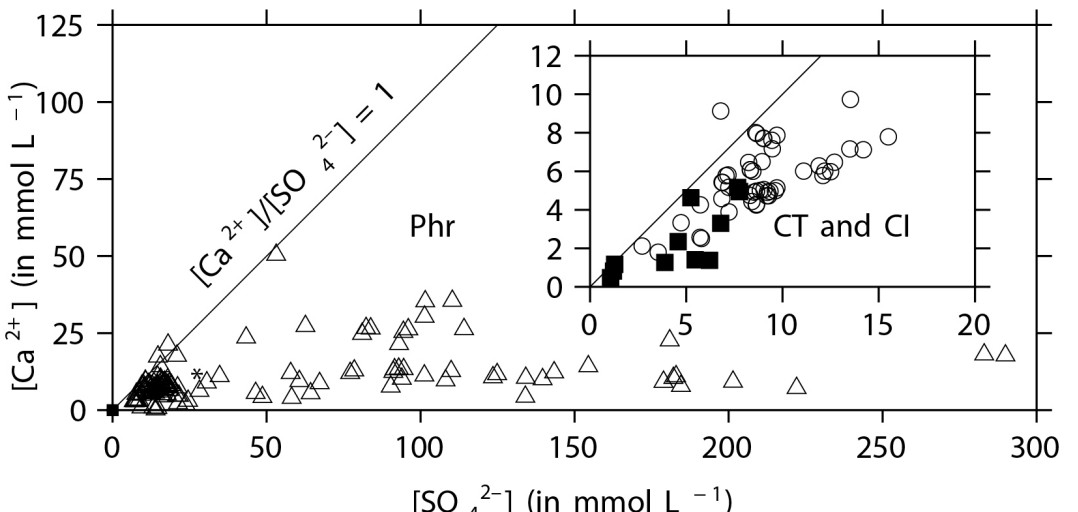

**Figure 14.** Calcium vs. $SO_4^{2-}$ diagram in Continental Intercalaire (filled squares), Complexe Terminal (open circles), Phreatic aquifer (open triangles) and Seawater composition (star) is $[Ca^{2+}] = 10.2 \, \text{mmol L}^{-1}$ and $[SO4^{2-}] = 28.2 \, \text{mmol L}^{-1}$ (Stumm and Morgan, 1999, p.899).

**HESSD**

doi:10.5194/hess-2015-385

**Chemical and isotopic data from groundwaters in Sahara**

R. Slimani et al.

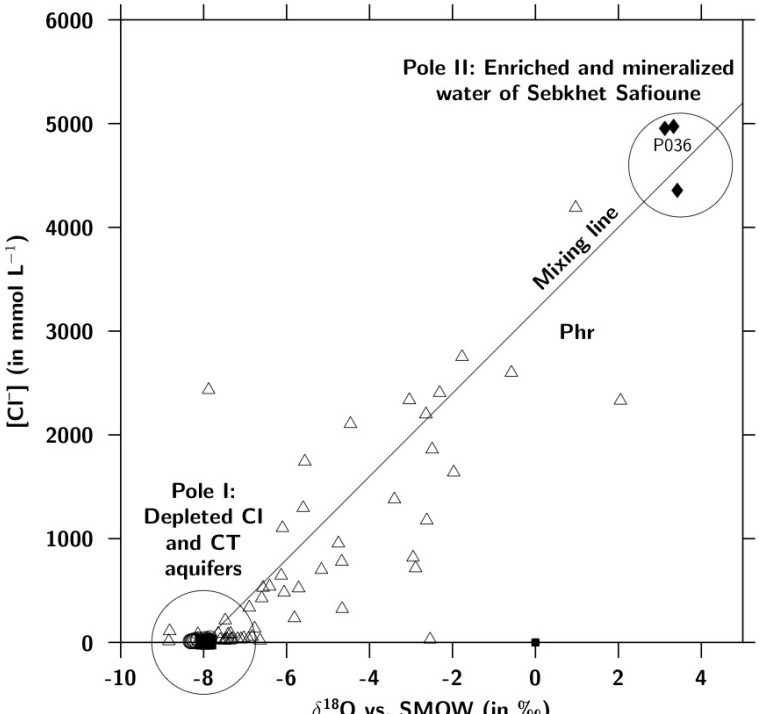

**Figure 15.** Chloride concentration vs. $\delta^{18}$O in Continental Intercalaire (filled squares), Complexe Terminal (open circles) and Phreatic aquifer (open triangles) from Ouargla.

Discussion Paper | Discussion Paper | Discussion Paper | Discussion Paper

**HESSD**

doi:10.5194/hess-2015-385

**Chemical and isotopic data from groundwaters in Sahara**

R. Slimani et al.

**Figure 16.** Log [Cl$^-$] concentration vs. $\delta^{18}$O in Continental Intercalaire (filled squares), Complexe Terminal (open circles) and Phreatic aquifer (open triangles) from Ouargla.

**HESSD**

doi:10.5194/hess-2015-385

**Chemical and isotopic data from groundwaters in Sahara**

R. Slimani et al.



**Figure 17.** Histogram of $Cl^-$ concentration in Phr samples from 1996 (Table 6) ($n = 59$). For 17 samples, $[Cl^-] < 35\,mmol\,L^{-1}$.