# Peer review of "Geochemical inverse modeling of chemical and isotopic data from groundwaters in Sahara (Ouargla basin, Algeria)"

_Hydrology and Earth System Sciences, 2015_

## Referee Comment (RC1) · G. Martinelli (Referee) · 23 Feb 2016

G. Martinelli (Referee)

giovannimartinelli@arpa.emr.it

Authors report about previous and new geochemical and isotopic data collected in the area of Ouargla, Algeria. The paper is well organized but deserves some improvements before to be published. Anyway some features are unclear and should be better described. In particular it's unclear by tab.6 when samples were collected. Probably they were collected in relatively recent times. Within this assumption I set up the enclosed table which shows that Phreatic samples are obviously more rich in Tritium while Complexe and Continental are relatively poor and consequently older. Authors report at pag.13 that precipitations were characterized by 16 TU in 1992. In what station? Anyway 1992 cannot reflect present day Tritium data and age evaluations should consider

more updated values. IAEA-GNIP network data report that Tritium was 6.8 in Algiers in 1999, 10.39 in Ankara in 2002 and 1.99 in Tenerife in 2010. Time series analysis of mentioned sites reveals that Tritium is still constantly lowering in precipitations. Stations are far from studied area anyway a Medioterranean-North African trend in isotopic data is recognizeable. It means that 16 TU is really an uncorrect value. More recent groundwaters are affected by a Tritium seasonal signal (phreatic-highest Tritium values), while samples characterized by 1-2 years of average permanence time could be suitable for a Tritium estimate which could be inferred by not too negative samples in 18O/16O (old groundwaters) and not too positive samples in 18O/16O (evaporation? Seasonal effects?) Thus values around 5-6 TU could be representative of present day rains (see also Martinelli et al., 2015, J.of Hydrology). More depleted waters in 18O should be old groundwaters characterized by a relatively remote recharge area, hopefully to be preserved. No Deuterium data available? Phreeqc approach to water composition is interesting anyway Authors should exclude by geological or stratigraphic data eventual mixing with brine waters. If the existence of brines is not excluded Phreeqc could be someway misleading.

---

## Referee Comment (RC2) · Anonymous Referee #2 · 12 Apr 2016

Understanding the reactive transport processes and parameters is very important for chemists, hydrologists and environmental scientists. This paper tries to interpret the geochemical measurements from different locations by inverse modeling, which is a standard numerical method for estimating geochemical parameters and simulating the fluid-rock interaction in heterogeneous aquifer systems. The results demonstrate that the geochemical reactions between minerals and fluids in saturated porous media are controlled by reactive mineral (e.g., gypsum and calcite) distributions and all Phr waters are resulted from the mixing of the two poles together with calcite precipitation and ion exchange processes. This is an improvement in the field-scale reactive chemical transport modeling.

[Figure]

The analytical results presented in this paper are convincing to me, and I have not seen these results being published elsewhere in literature. The work presented in this paper will have a solid contribution to the study of the reactive chemical transport in field-scale porous media. The scientific community (especially people interested in reactive chemical transport modeling, environmental quality assessment, geochemical parameter characterization, and multiple-scale stochastic modeling) may find interest in the analysis and results of this manuscript.

The reviewer suggests the manuscript be accepted for publication after some minor revisions:

1. The authors used lots of "bullets" in the manuscript to describe the modeling works. This kind of descriptions is usually used in technical reports other than an academic paper. I am wondering if the authors can incorporate them in a paragraph?

2. In "Introduction", the authors should expend the literature review to include more recently published papers relevant to reactive chemical transport modeling. You may conduct a Google Search of "inverse modeling of reactive transport" to find the recently published papers such as (Dai and Samper, 2004).

3. The authors may need more explanation about what measured geochemical data and how these data are used for inverse modeling.

4. In "Conclusion", the authors may discuss more about the scientific importance of this study, which may strengthen the conclusions and this paper.

---

## Author Comment (AC2) · 11 May 2016

1. The bullets that are used in the manuscript will be reduced as much as possible especially with regard to the inverse modeling section.

2. The literature review will be expanded to include more references of recently published papers such as those suggested (Dai & Sampers, 2004) and a couple of others.

3. All geochemical data that was measured is presented in the tables. As the authors used PhreeqC, the methodology is already constrained by its use and the mineral that are known and defined by the geological and lithological features of the region of interest has done the rest in the selection of the data to be used

4. The suggestion of highlighting the scientific importance of inverse modelling has been taken into consideration

---

## Author Response (AR1)

Dear editor,

The authors are very thankful and grateful to the two referees who accepted to evaluate the work presented in the submitted manuscript and for their valuable remarks.

**Answers to Referee #1 (G. Martinelli) comments:**

Values displayed in table 6 are dated back to November 1992 so they are old values. This is the main reason why they are considered high comparatively to what is expected to be found nowadays. In fact, at present times, tritium figures have fallen lower than 5 TU in precipitation measured in the northern part of the country.

Where and when the rain sample was collected:
The value of 16 TU in precipitation was collected from Ouargla itself (from the National Agency for Water Resources (ANRH) station).

The value of 16 TU seems to be high but we can note the following remarks:

We are in an arid area (desert) where precipitation is very scarce and irregular. Precipitation takes place in the form of sudden thunderstorms in and unsaturated atmosphere and a great part of this precipitation evaporates back into the moisture unsaturated atmosphere sometimes during many cycles. Consequently, an enrichment in tritium happens because when water evaporates back, the lightest fractions (isotopes) are the ones that evaporate first causing an enrichment in Tritium in the remaining fraction. The 16 TU value would thus correspond then to a rainy event that had happened during the same sampling period (Nov. 1992). It's the only available value and it's is not a weighted mean for a long period of time. It's the most representative value for that region and for that time. Unfortunately, all the other stations (Algiers, Ankara, and Tenerife) are subject to a completely different climatic regime and beside the fact that they have more recent values, can absolutely not be used for our case. Therefore all the assumptions based on recent tritium rain values do not apply to this study.

Unfortunately due to a technical problem at that time, no deuterium values were made available for those samples

Depleted contents in O-18 and low tritium concentrations for phreatic waters fit well the mixing scheme and confirm the contribution from the older and deeper CI/CT groundwaters. The affected areas were clearly identified in the field and correspond to locations that are subject to a recycling and a return of irrigation waters whose origin are CI/CT boreholes. Moreover, the mixing that is clearly brought to light by the Cl vs. O-18 diagrams (Fig. 15 & 16) could partly derive from an ascending drainage from the deep and confined CI aquifer (exhibiting depleted homogenous O-18 contents & very low tritium), a vertical leakage that is favoured by the Amguid El-biod highly faulted area (geological argument).

***Answers to Referee #2 (Anonymous) comments:***

1- The bullets that are used in the manuscript will be reduced as much as possible especially with regard to the inverse modeling section.

We assume that the relationship between $^{18}$O and Cl data obtained in 1996 is stable with time, which is a logical assumption as times of transfer from CI to both CT and Phr are very long. Considering both $^{18}$O and Cl data, CI, CT and Phr data populations can be categorized. The CI and CT do not show appreciable $^{18}$O variations, and can be considered as a single population. The Phr samples consist however of different populations: Pole I, with $\delta^{18}$O values close to -8, and small Cl concentrations, more specifically less than 35 mmol.l$^{-1}$; Pole II, with $\delta^{18}$O values larger than 3, and very large Cl concentrations, more specifically larger than 4000 mmol.l$^{-1}$; intermediate Phr samples result from mixing between poles I and II (mixing line in Fig. 13, mixing curve in Fig. 14) and from evaporation of pole I (evaporation line in Fig. 14).

2- The literature review will be expanded to include more references of recently published papers such as those suggested (Dai & Sampers, 2004) and a couple of others.

In the present study, new data were collected in order to characterize the hydrochemical and the isotopic composition of the major aquifers in Ouargla's region. They also aimed at identifying the origin of the mineralization and water-rock interactions that occur along the flow. New possibilities offered by progress in geochemical simulations were used. More specifically, the inverse modeling of chemical reactions allows us to select the best conceptual model for the interpretation of the geochemical evolution of the Ouargla aquifer. The stepwise inversion strategy involves designing a list of the scenarios that includes the most plausible combinations of geochemical processes, solving scenarios in a stepwise manner, and selecting the scenario that provides the best conceptual geochemical model (Dai et al., 2006). Inverse modeling with Phreeqc 3.0 was used to quantitatively assess the influence of the processes that explain the acquisition of solutes for the different aquifers: dissolution, precipitation, mixing and ion exchange. This results in constraints on mass balances as well as on the exchange of matter between aquifers.

3- All geochemical data that was measured is presented in the tables. As the authors used PhreeqC, the methodology is already constrained by its use and the mineral that are known and defined by the geological and lithological features of the region of interest has done the rest in the selection of the data to be used

[revised manuscript text omitted]
 | P416 | 3,538,292.9 | 719,746 | 132 | 27/01/2013 | 5.51 | 24.6 | 8.44 | 2.37 | 38.8 | 16.9 | 36.9 | 1.93 | 9.03 | 9.21 | |
| PARC SONACOM | PL10 | 3,556,077 | 720,442.9 | 134 | 21/01/2013 | 6.09 | 25.4 | 7.78 | 1.62 | 39.7 | 11.7 | 36 | 8.43 | 5.11 | 5.97 | |
| Bour El Haicha | LTPSN2 | 3,544,999.3 | 719,558 | 110 | 27/01/2013 | 6.08 | 24.5 | 8.13 | 1.82 | 39.8 | 11.8 | 30.6 | 5.2 | 7.14 | 8.46 | |
| Route Ain Moussa | PL28 | 3,549,933 | 719,930.6 | 128 | 21/01/2013 | 6.22 | 23.1 | 8.07 | 1.82 | 42 | 19.1 | 27.5 | 13.21 | 13.39 | 8.12 | |
| Route Ain Moussa | P408 | 3,549,933 | 717,022 | 128 | 1996 | 7.62 | 23.65 | 7.93 | 0.56 | 42.14 | 10.72 | 18.87 | 1.86 | 12.63 | 9.32 | |
| Ecole Okba B. Nafaa | P056 | 3,549,933 | 717,022 | 128 | 26/01/2013 | 5.98 | 24.6 | 7.63 | 2.16 | 42.5 | 17.9 | 32.1 | 8.03 | 12.49 | 8.07 | |

[revised manuscript text omitted]

---

## Author Response (AR2)

[revised manuscript text omitted]
 | PL.23 | 3,538.419 | 720.950 | 126 | 1992 | | 23.49 | 7.37 | 3 | 1,103.31 | 94.49 | 707.81 | 19.14 | 270.91 | 13.3 |
| Sebkhet Safioune | P023 | 3,577.198 | 725.726 | 99 | 1992 | | 23.32 | 7.42 | 2.25 | 1,176.99 | 91.14 | 1,058.21 | 11.72 | 133.47 | 12.41 |
| Sebkhet Safioune | P034 | 3,579.698 | 725.633 | 97 | 05/02/2013 | 130 | 34.9 | 8.08 | 1.76 | 1,189.1 | 14.7 | 1,055.1 | 18.27 | 56.37 | 17.38 |
| Chott Adjadja | PLXI | 3,540.758.8 | 726.115.6 | 132 | 1992 | 117.9 | 29.4 | 8.19 | 1.85 | 1,209.3 | 15.6 | 1,129.4 | 8.38 | 42.85 | 10.15 |
| Sebkhet Safioune | P063 | 3,545.586.8 | 725.667.4 | 99 | 1992 | | 23.60 | 8.02 | 3.82 | 1,296.65 | 134.01 | 1,458.73 | 5.24 | 47.98 | 4.34 |
| Sebkhet Safioune | LTP06 | | | | 1992 | | 23.50 | 7.46 | 1.94 | 1,379.35 | 139.61 | 1,257.42 | 18.6 | 182.26 | 10.03 |
| Bamendil | P076 | 3,540.137 | 716.721 | 118 | 1992 | | 23.77 | 7.64 | 7.84 | 1,638.66 | 712.09 | 2,621.61 | 41.55 | 190.51 | 13.34 |
| El Bour-N'goura | P007 | 3,562.236 | 718.651 | 129 | 1992 | | 23.53 | 7.71 | 5.72 | 1,743.65 | 143.36 | 1,321.87 | 26.85 | 331.38 | 12.26 |
| Sebkhet Safioune | P063 | 3,545.586.8 | 725.667.4 | 99 | 05/02/2013 | 178.9 | 23.26 | 7.67 | 1.41 | 1,860.53 | 91.55 | 1,434.73 | 26.2 | 278.77 | 13.25 |
| | P044 | | | | 1992 | | 26.7 | 7.67 | 1.43 | 1,887.9 | 92.9 | 1,455.8 | 26.66 | 282.88 | 13.44 |
| | P093 | | | | 1992 | | 23.39 | 7.79 | 4.53 | 2,106.07 | 18.27 | 1,765.47 | 27.33 | 171.23 | 6.54 |
| | P042 | | | | 1992 | | 23.58 | 7.49 | 1.49 | 2,196.58 | 182.08 | 1,957.53 | 29.49 | 278.18 | 10.44 |
| | P093 | | | | 1992 | | 23.42 | 7.59 | 1.1 | 2,330.85 | 101.22 | 1,963.71 | 52.19 | 248.1 | 11.24 |
| | P068 | | | | 1992 | | 23.51 | 7.54 | 3.35 | 2,335.67 | 222.08 | 2,302.25 | 26.84 | 219.9 | 7.19 |
| Oum Raneb | PZ12 | 3,547.234 | 722.931 | 110 | 1992 | | 23.31 | 7.59 | 2.21 | 2,405.55 | 222.08 | 2,178.55 | 25.23 | 199.35 | 12.65 |
| Hassi Debich | P416 | 3,581.097 | 730.922 | 106 | 1992 | | 23.34 | 7.84 | 4.33 | 2,433.73 | 178.87 | 2,361.09 | 24.34 | 196.07 | 9.2 |
| N'Goussa | P041 | 3,559.563 | 716.543 | 135 | 1992 | | 23.38 | 7.94 | 2.13 | 2,599.74 | 324.58 | 2,878.99 | 44.57 | 152.83 | 10.97 |
| Sebkhet Safioune | P034 | 3,579.698 | 725.633 | 97 | 1992 | | 23.34 | 7.85 | 1.95 | 2,752 | 134.14 | 2,616.77 | 24.42 | 180.14 | 10.48 |
| | P039 | | | | 1992 | | 23.37 | 6.87 | 1.94 | 4,189.51 | 201.44 | 4,042.62 | 17.9 | 257.81 | 9.23 |
| Sebkhet Safioune | P074 | | | | 1992 | | 23.54 | 6.47 | 4.17 | 4,356.48 | 180.88 | 2,759.9 | 57.4 | 930.06 | 22.63 |
| Sebkhet Safioune | P037 | | | | 1992 | | 23.36 | 6.92 | 1.52 | 4,953.84 | 184.54 | 4,611.06 | 2.9 | 347.57 | 7.86 |
| Sebkhet Safioune | P036 | | | | 1992 | | 23.35 | 7.54 | 1.4 | 4,972.75 | 108.12 | 4,692.23 | 36.84 | 221.13 | 9.63 |

[revised manuscript text omitted]

Dear editor,

The following modifications have been made:

1- The address has been modified
- Line 4 : [a]Univ Ouargla, Fac. des sciences….
- [a]Ouargla University, Fac. des Sciences de la Nature et de la Vie, Lab. Biochimie des Milieux Désertiques, Ouargla 30000, Algeria.

2- The address has been modified
- Line 8 : [d]Nuclear Research Centre….
- [d]Algiers Nuclear Research Centre, P.O. Box, 399 Alger-RP, 16000 Algiers, Algeria.

3- The abstract has been modified accordingly
- Line 10 : New samples….
- Unpublished chemical and isotopic data taken in November 1992 from the three major Saharan aquifers namely, the Continental Intercalaire (CI), the Complexe Terminal (CT) and the Phreatic aquifer (Phr) were integrated with original samples in order to chemically and isotopically characterize a Saharan aquifer system and investigate the processes through which groundwaters acquire their mineralization.

4- What do you mean pole?
- Line 17 : …a first pole of Phr….
- The pole mean «end member, clustere»

5- The Reference added
- Line 25 : A scientific study…
- A scientific study published in 2008 (OECD, 2008).

6- It has been replaced
- Line 29 : …causing most of the time overuse…
- often causing overuse.

7- It has been replaced
- Line 33 : … is almost unexploited (only north of Ouargla) due…
- is almost unexploited, due to its salinity (50 g/L);

8- It has been deleted
- Line 34 : … "Complexe Terminal" (CT) aquifer,…
- in the middle, the "Complexe Terminal" (CT)

9- It has been deleted
- Line 35 : …which is the most exploited, and includes…
- is the most exploited and includes

10- It has been replaced
- Line 37 : …Miopliocene…

‒ Mio-pliocene

11- It has been replaced

• Line 37 : …"Continental Intercalaire" (CI) aquifer, where water is contained in…

[revised manuscript text omitted]

24- It has been rephrased

- Line 157 : The salinity of the Complexe Terminal…
- The salinity of the CT (Mio-pliocene) aquifer (Fig. 5b) is much lower than that of the Phr aquifer, and ranges from 1 to 2 g/L;

25- It has been rephrased

- Line 177 : No significant…
- No significant saturation indices' evolution from the south to the north upstream and downstream of Oued Mya.

26- It has been rephrased

- Line 193 : For most of the sampled…
- [Na$^+$]/[Cl$^-$] ratio is from 0:85 to 1:26 for CI aquifer, from 0:40 to 1:02 for the CT aquifer and from 0:13 to 2:15 for the Phr aquifer.

27- I wonder whether it could be useful to add this line to the plots of figure 10

- Line 202 :…the seawater mole ratio (0,858), …
- There is a star * in the plot, and the values are given in the caption of figure 10, but the values are very close to the 1:1 line and masked by samples.

28- It has been rephrased

- Line 209 : In these aquifers,…
- In the CI, CT and Phr aquifers, calcium originates both from carbonate and sulfate (Fig. 11 and 12). Three samples from CI aquifer are close to the [Ca2+]/[HCO3–] 1:2 line, while calcium sulfate dissolution explains the excess of calcium. However, nine samples from phreatic aquifer are depleted in calcium, and plot under the [Ca2+]/[HCO3–] 1:2 line.

29- Recall whether this linement has a geological or hydrogeological importance.

- Line 227 : Waters located north of the Hassi Miloud to Sebkhet Safioune axis are more enriched in heavy isotopes and therefore more evaporated.
- This is not a linement of hydrogeological importance, but results from anthropogenic influence by irrigation. Far from Ouargla, there is no irrigation, while in the vicinity of Ouargla, irrigation waters are directly pumped in the CI and mostly CT aquifers, so these irrigation waters both evaporate and mix with Phr waters.

30- Symbol changed to constant

- Line 249 : Equation 1

31- It has been rephrased and the order of sentences modified

- Line 254 : There is only one sample…

 P115 is the only sample that appears on the straight evaporation line (Fig. 14). It should be considered as an outlier since the rest of the samples are all well alined on the logarithmic fit derived from the mixing line of Figure 13.

32- Equation 3 has been changed

- Line 266 : $\delta_{mix} = f_1 \times \delta_1 + f_2 \times \delta_2$
– $\delta_{mix} = f_1 \times \delta_1 + (f_1 - 1) \times \delta_2$

33- It has been rephrased

- Line 296 : This values are dated…
– These values are dated back to November 1992 so they are old values and they are considered high comparatively to what is expected to be found nowadays.

34- It has been rephrased.

- Line 290 : The comparison of these…
– These values are dated back to November 1992 so they are old values and they are considered high comparatively to what is expected to be found nowadays.

35- The whole paragraph has been modified

- Line 292 : This value seems…
– Tritium content of precipitation was measured as 16 TU in 1992 on a single sample that was collected from the National Agency for Water Resources station in Ouargla. A major part of this raifall evaporates back into the atmosphere that is unsaturated in moisture. Consequently, enrichment in tritium happens as water evaporates back.

36- It has been rephrased.

- Line 354 : In a decreasing order ...
– In a descending order of amount, halite, sylvite, gypsum and huntite are the minerals that are the most involved in the dissolution process.

37- Reference added in the tables

- Tableau : In these tables, provide information about the reference system for Latitude and Longitude. Moreover, some data are given with decimal digits. Is this physically significant?
– The reference is UTM 31 projection for North Sahara 1959 (CLARKE 1880 ellipsoid). The decimal digits are not physically significant, and simply indicative to locate sampling sites.

Best regards

*Slimani Rabia*

---

## Editor Decision (ED2)

[revised manuscript text omitted]

Dear editor,

The following modifications have been made:

1- The address has been modified
- Line 4 : [a]Univ Ouargla, Fac. des sciences….
- [a]Ouargla University, Fac. des Sciences de la Nature et de la Vie, Lab. Biochimie des Milieux Désertiques, Ouargla 30000, Algeria.

2- The address has been modified
- Line 8 : [d]Nuclear Research Centre….
- [d]Algiers Nuclear Research Centre, P.O. Box, 399 Alger-RP, 16000 Algiers, Algeria.

3- The abstract has been modified accordingly
- Line 10 : New samples….
- Unpublished chemical and isotopic data taken in November 1992 from the three major Saharan aquifers namely, the Continental Intercalaire (CI), the Complexe Terminal (CT) and the Phreatic aquifer (Phr) were integrated with original samples in order to chemically and isotopically characterize a Saharan aquifer system and investigate the processes through which groundwaters acquire their mineralization.

4- What do you mean pole?
- Line 17 : …a first pole of Phr….
- The pole mean «end member, clustere»

5- The Reference added
- Line 25 : A scientific study…
- A scientific study published in 2008 (OECD, 2008).

6- It has been replaced
- Line 29 : …causing most of the time overuse…
- often causing overuse.

7- It has been replaced
- Line 33 : … is almost unexploited (only north of Ouargla) due…
- is almost unexploited, due to its salinity (50 g/L);

8- It has been deleted
- Line 34 : … "Complexe Terminal" (CT) aquifer,…
- in the middle, the "Complexe Terminal" (CT)

9- It has been deleted
- Line 35 : …which is the most exploited, and includes…
- is the most exploited and includes

10- It has been replaced
- Line 37 : …Miopliocene…

- Mio-pliocene

11- It has been replaced

- Line 37 : …"Continental Intercalaire" (CI) aquifer, where water is contained in…
- at the bottom, the "Continental Intercalaire" (CI), hosted in the lower Cretaceous

12- it has been rephrased

- Line 46 : tried, starting from …………Continental Intercalaire recharge.
- started from chemical and isotopic information ($^2$H, $^{18}$O, $^{234}$U, $^{238}$U, $^{36}$Cl) to characterize the relationships between aquifers. In particular, such studies focused on the recharge of the deep CI aquifer system.

13- Stress the scientific novelty of this paper. From this descriptio it seems that nothing really new is proposed from the scientific point of view, but one of the referees stressed the importance of reducing the lack of data for african regions. I think you can reinforce both the methodlogical aspects (which is the open scientific question? How is it faced in this work?) and the "geographical" application.

- Line 53 : In the present study…
- The present study aims at applying for the first time ever in Algeria, a new methodology (inverse modeling) to an extreme environment where lack of data on a scarce natural resource (groundwater) is observed. New data were hence collected in order to characterize the hydro chemical and the isotopic composition of the major aquifers in the Saharan region of Ouargla.

14- The word has been replaced

- Line 56 : …simulations…
- …modeling…

15- it is now explained

- Line 60 : The stepwise…
- The stepwise inversion strategy involves designing a list of the scenarios that includes the most plausible combinations of geochemical processes, solving scenarios in a stepwise manner, and selecting the scenario that provides the best conceptual geochemical model.

16- It has been deleted

- Line 69 : the quaternary fossil valley of OuedMya basin
- It is located in the quaternary valley of Oued Mya basin.

17-The sentences have been modified and completed

- Line 75 : During Secondary era…
- During Secondary era, vertical movements a_ected the Precambrian basement causing in particular collapse of its central part, along an axis passing approximately through the Oued Righ valley and the upper portion of the valley ouedMya. According to Furon (1960), a epicontinental sea spread to the Lower Eocene of northern Sahara. After the Oligocene, the sea gradually withdrew.

18- It has been rephrased

- Line 100 : The sampling scheme…

− The sampling scheme complies with the flow directions of the two formations (Phr and CT aquifers); for the CI aquifer only five points are available, so it is impossible to choose a transect (Fig. 3). Groundwater samples (n = 107) were collected during a field campaign in 2013, along the main flow line of Oued Mya, 67 piezometers tap the phreatic aquifer, 32 wells tap the CT aquifer and 8 boreholes tap the CI aquifer (Fig. 3).

19- The Reference added

- Line 112 : Scatchard and Guggenheim….

− Phreeqc was used to check minerals / solution equilibria using the specific interaction theory (SIT), i.e. the extension of Debye-Hückel law by Scatchard and Guggenheim incorporated recently in Phreeqc 3.0 (Parkhurst and Appelo, 2013).

20- The whole paragraph has been modified

- Line 119 : The Inverse modeling……

− Inverse modeling involves designing a list of scenarios (modelling setups) that take into account the most plausible combinations of geochemical processes that are likely to occur in our system. For example, the way to identify whether calcite dissolution/precipitation is relevant or not consists of solving the inverse problem under two alternate scenarios: (1) considering a geochemical system in which calcite is present, and (2) considering a geochemical system without calcite. After simulating the two scenarios, it is usually possible to select the setup that gives the best results as the solution to the inverse modeling according to the fit between the modeled and observed values.

21- It has been rephrased

- Line 129 : Samples are…

− Samples are ordered according to an increasing electric conductivity (EC), and this is assumed to provide an ordering for increasing salt content. In both phreatic and CT aquifers, temperature is close to 25 °C, while for CI aquifer, temperature is close to 50 °C. The values presented in tables 1 to 5 are raw analytical data that were corrected for defects of charge balance before computing activities with Phreeqc.

22- It has been rephrased

- Line 141 : The facies of the Phreatic aquifer …

− Respectively, $CaSO_4$, $Na_2SO_4$, $MgSO_4$ and $NaCl$ are the most dominant chemical species (minerals) that are present in the phreatic waters. This sequential order of solutes is comparable to  that of other groundwater occurring in North Africa, and especially in the neighboring area of the chotts (depressions where salts concentrate by evaporation) Merouane and Melrhir.

23- It has been replaced

- Line 148 : The salinity of the phreatic aquifer...

- The salinity of the phreatic aquifer varies considerably depending on the location (namely, the distance from wells or drains) and time (due to the influence of irrigation) (Fig. 5a).

24- It has been rephrased

- Line 157 : The salinity of the Complexe Terminal…
- The salinity of the CT (Mio-pliocene) aquifer (Fig. 5b) is much lower than that of the Phr aquifer, and ranges from 1 to 2 g/L;

25- It has been rephrased

- Line 177 : No significant…
- No significant saturation indices' evolution from the south to the north upstream and downstream of Oued Mya.

26- It has been rephrased

- Line 193 : For most of the sampled…
- [Na$^+$]/[Cl$^-$] ratio is from 0:85 to 1:26 for CI aquifer, from 0:40 to 1:02 for the CT aquifer  and from 0:13 to 2:15 for the Phr aquifer.

27- I wonder whether it could be useful to add this line to the plots of figure 10

- Line 202 :…the seawater mole ratio (0,858), …
- There is a star * in the plot, and the values are given in the caption of figure 10, but the values are very close to the 1:1 line and masked by samples.

28- It has been rephrased

- Line 209 : In these aquifers,…
- In the CI, CT and Phr aquifers, calcium originates both from carbonate and sulfate (Fig. 11 and 12). Three samples from CI aquifer are close to the [Ca2+]/[HCO3–] 1:2 line, while calcium sulfate dissolution explains the excess of calcium. However, nine samples from phreatic aquifer are depleted in calcium, and plot under the [Ca2+]/[HCO3–] 1:2 line.

29- Recall whether this linement has a geological or hydrogeological importance.

- Line 227 : Waters located north of the Hassi Miloud to Sebkhet Safioune axis are more enriched in heavy isotopes and therefore more evaporated.
- This is not a linement of hydrogeological importance, but results from anthropogenic influence by irrigation. Far from Ouargla, there is no irrigation, while in the vicinity of Ouargla, irrigation waters are directly pumped in the CI and mostly CT aquifers, so these irrigation waters both evaporate and mix with Phr waters.

30- Symbol changed to constant

- Line 249 : Equation 1

31- It has been rephrased and the order of sentences modified

- Line 254 : There is only one sample…

 P115 is the only sample that appears on the straight evaporation line (Fig. 14). It should be considered as an outlier since the rest of the samples are all well alined on the logarithmic fit derived from the mixing line of Figure 13.

32- Equation 3 has been changed

- Line 266 : $\delta_{mix} = f_1 \times \delta_1 + f_2 \times \delta_2$
- $\delta_{mix} = f_1 \times \delta_1 + (f_1 - 1) \times \delta_2$

33- It has been rephrased

- Line 296 : This values are dated…
- These values are dated back to November 1992 so they are old values and they are considered high comparatively to what is expected to be found nowadays.

34- It has been rephrased.

- Line 290 : The comparison of these…
- These values are dated back to November 1992 so they are old values and they are considered high comparatively to what is expected to be found nowadays.

35- The whole paragraph has been modified

- Line 292 : This value seems…
- Tritium content of precipitation was measured as 16 TU in 1992 on a single sample that was collected from the National Agency for Water Resources station in Ouargla. A major part of this raifall evaporates back into the atmosphere that is unsaturated in moisture. Consequently, enrichment in tritium happens as water evaporates back.

36- It has been rephrased.

- Line 354 : In a decreasing order ...
- In a descending order of amount, halite, sylvite, gypsum and huntite are the minerals that are the most involved in the dissolution process.

37- Reference added in the tables

- Tableau : In these tables, provide information about the reference system for Latitude and Longitude. Moreover, some data are given with decimal digits. Is this physically significant?
- The reference is UTM 31 projection for North Sahara 1959 (CLARKE 1880 ellipsoid). The decimal digits are not physically significant, and simply indicative to locate sampling sites.

Best regards

*Slimani Rabia*

---

## Author Response (AR3)

Dear editor,

The authors are very thankful and grateful to Associate Editor for their valuable remarks.

The following modifications have been made:

1- The title has been modified

- Line 1 : Geochemical inverse modeling of chemical and isotopic data from groundwaters in Sahara (Ouargla basin, Algeria).
– Identification of dominant hydrogeochemical processes for groundwaters in Algerian Sahara supported by inverse modeling of chemical and isotopic data.

2- The whole paragraph has been modified

- Lines 40 to 55 : The present study…
– The present study aims at applying for the first time ever in Algeria, inverse modeling to an extreme environment, characterized by a lack of data on a scarce natural resource (groundwater). In the present study, new data were collected in order to characterize the hydrochemical and the isotopic composition of the major aquifers in Ouargla's region and identify the origin of the mineralization and water-rock interactions that occur along the flow. New possibilities offered by progress in geochemical simulations were used. More specifically, evaporite dissolution, ion exchange, calcite dissolution / precipitation and $CO_2$ escape or dissolution and mixing can be quantitatively assessed by inverse modeling (Dai and al., 2006) with Phreeqc 3.0 to explain the modifications of the chemical composition of the three main Saharan aquifers. This results in constraints on mass balances as well as on the exchange of matter between aquifers.

3- It has been rephrased.

- Line 83 : The exploitation of
– The exploitation of Senonian aquifer dates back to 1953 at a depth between 140 to 200 m,

4- It has been deleted.

- Line 92 : A total of (n = 107) samples…
– A total of 107 samples were collected during a field campaign in 2013,

5- It has been replaced.

- Line 177 : This is illustrated…
– This is illustrated by a decrease of the $[HCO_3^-]/ ([Cl^-] + 2[SO_4^{2-}])$ ratio (Fig. 8) from 0.2 to 0 and of the $[SO_4^{2-}]/[Cl^-]$ ratio from 0.8 to values smaller than 0.3 (Fig. 9) while salinity increases.

6- It has been replaced.

- Line 185 : $[Na^+]/[Cl^-]$ ratio is from…
– $[Na^+]/[Cl^-]$ ratio ranges from 0.85 to 1.26 for CI aquifer, between 0.40 and 1.02 for the CT aquifer, between 0.13 and 2.15 for the Phr aquifer. The measured points from the three

considered aquifers are linearly scattered with good approximation around the unity slope straight line that stands for halite dissolution (Fig. 10).

7- It has been replaced.

- Line 203 : However, nine samples from phreatic aquifer
− However, nine samples from Phr aquifer

8- It has been corrected

- Line 243 : factor and K is a constant
− factor and $k$ is a constant

9- It has been replaced.

- Line 268 : …(cf. infra. 3.6.).
− …(see section 3.6)

10- It has been corrected

- Line 285 : …this raifall…
− A major part of this rainfall

11- It has been replaced.

- Equations 6, 10 and 11: …+ cte

− …+ *constant*

12- It has been corrected.

- Line 430 : …   = 100( $\alpha$ - 1)
− …   = 1000( $\alpha$ - 1)

13- The decimals have been erased from latitude and longitude values.

14- We kept one significant digit more than in table 7 to avoid rounding errors if anyone wants to reuse our data.

15- The figures 8 and 9 have been improved by using log-log axes.

16- In figures 11 and 12, the square at the origin of the axes is not corresponding to a sample belonging to CI.

17- The point of Seawater has been added in figures 11 and 12

Best regards

*SLIMANI Rabia*

---

## Author Response (AR4)

UMR 1114 Emmah

Pr. Mauro Giudici
Editor HESS

Avignon, Ouargla, October 31th 2016

Dear Professor Giudici,

thank you very much for your comments and suggestions.

We modified our manuscript to show how it can be of interest to an international audience:

- the aquifers studied are important both quantitatively and as a scarce resource in a semi-arid to hyper-arid environment; there are very few studies on these aquifers;

- inverse modeling is used to quantitatively assess the modifications of the chemical composition of water along the flowpath combining mass transfer between aquifers and water and reactive mixing; isotope data are used to discriminate statistically equivalent models;

- the methodology and data can be integrated in a straightforward manner in a general framework where geochemical models (PhreeqC or others), crop models and land use models are operationally interfaced, as the achievement of cooperative efforts during the last decades referred to at the end of the introduction.

Modifcations are not limited to the introduction and are shown by correction marks in the margins throughout the manuscript.

With our best regards,

Pr. Guilhem Bourri and Dr Rabia Slimani

Institut National de la Recherche Agronomique
Domaine St Paul, Site Agroparc, F84914 Avignon cedex 09 France
Tl.: +33 (0)4 32 72 22 28 Fax : + 33 (0)4 32 72 22 12 Courriel:
guilhem.bourrie@inra.fr

[revised manuscript text omitted]

---

## Author Response (AR5)

Pr. Mauro Giudici

Editor HESS

Avignon, Ouargla, November 28[th] 2016

Dear Professor Giudici,

The authors are very thankful and grateful for this valuable remarks.

The following modifications have been made:

1- The word "Captive" in line 16, signified confined.

2- The whole paragraph has been rephrase.

• Lines 19 to 27 : This system covers…

− This system covers a surface of more than one million $km^2$ (700,000 $km^2$ in Algeria, 80,000 $km^2$ in Tunisia and 250,000 $km^2$ in Libya). Due to the climatic conditions of Sahara, these formations are poorly renewed: about 1 billion $m^3$/year essentially infiltrated in the Piedmont of the Saharan Atlas in Algeria, as well as in the Dahar and Djebel Nafusa in Tunisia and Libya respectively. However, the very large extension of the system as well as the great thickness of the aquifer layers has favored the accumulation of huge water reserves. Ouargla basin is located in the middle of the NWSAS and thus benefits from groundwater resources (Fig. 1) which are contained in the following three main reservoirs (UNESCO, 1972; Eckstein and Eckstein, 2003; OSS, 2003, 2008).

3- It has been rephrased.

• Lines 407 to 408 : From the three aquifers...

− Two of the aquifers studeied in this work, Complexe Terminal and Continental Intercalaire,...

4- It has been rephrased.

• Lines 409 to 410 : The last one...

− The last one, Phreatic aquifer, is a shallow aquifer. The chemical facies of these aquifers have long been qualitatively described.

5- The measurements units has been added on the axes in figures 3 and 5.

With our best regards,

Pr. Guilhem Bourri and Dr Rabia Slimani.